# CEA-CD3 bispecific antibody cibisatamab with or without atezolizumab in patients with CEA-positive solid tumours: results of two multi-institutional Phase 1 trials

Neil H. Segal [1] ✉, Ignacio Melero [2,3], Victor Moreno [4], Neeltje Steeghs [5], Aurelien Marabelle [6], Kristoffer Rohrberg [7], Maria E. Rodriguez-Ruiz[2], Joseph P. Eder[8], Cathy Eng[9], Gulam A. Manji[10], Daniel Waterkamp[11], Barbara Leutgeb[12], Said Bouseida [12], Nick Flinn[12], Meghna Das Thakur[11], Markus C. Elze[12], Hartmut Koeppen[11], Candice Jamois[12], Meret Martin-Facklam[12], Christopher H. Lieu [13], Emiliano Calvo [14], Luis Paz-Ares[15], Josep Tabernero [16] & Guillem Argilés[16,17]

Cibisatamab is a bispecific antibody-based construct targeting carcinoembryonic antigen (CEA) on tumour cells and CD3 epsilon chain as a T-cell engager. Here we evaluated cibisatamab for advanced CEA-positive solid tumours in two open-label Phase 1 dose-escalation and -expansion studies: as a single agent with or without obinutuzumab in S1 (NCT02324257) and with atezolizumab in S2 (NCT02650713). Primary endpoints were safety, dose finding, and pharmacokinetics in S1; safety and dose finding in S2. Secondary endpoints were anti-tumour activity (including overall response rate, ORR) and pharmacodynamics in S1; anti-tumour activity, pharmacodynamics and pharmacokinetics in S2. S1 and S2 enrolled a total of 149 and 228 patients, respectively. Grade ≥3 cibisatamab-related adverse events occurred in 36% of S1 and 49% of S2 patients. The ORR was 4% in S1 and 7% in S2. In S2, patients with microsatellite stable colorectal carcinoma (MSS-CRC) given flat doses of cibisatamab and atezolizumab demonstrated an ORR of 14%. In S1 and S2, 40% and 52% of patients, respectively, developed persistent anti-drug antibodies (ADAs). ADA appearance could be mitigated by obinutuzumab-pretreatment, with 8% of patients having persistent ADAs. Overall, cibisatamab warrants further exploration in immunotherapy combination strategies for MSS-CRC.

New treatment options aim to expand the benefit of cancer immunotherapy beyond inflamed tumours[1]. Despite the approval of immune checkpoint inhibitors in a diverse range of solid malignancies[2], these therapies lack efficacy in the majority of patients. This is particularly true for patients with cancers that are characterized as nonimmunogenic (sometimes termed cold)[3] because they lack sufficient tumour-specific T cells, have insufficient expression of neoantigens, show defects in major histocompatibility complex antigen-presentation machinery and/or are rich in immunosuppressive factors in the tumour microenvironment[2,4–7]. Bispecific T-cell engagers were developed to redirect cytotoxic T cells to predefined tumour targets, primarily

for major histocompatibility complex–independent cancer cell elimination[8].

Cibisatamab is a T-cell bispecific antibody (TCB) that uses a 2-to-1 molecular format to target carcinoembryonic antigen (CEA) expressed on the surface of tumour cells and CD3ε on T cells[9]. A human immunoglobulin G1 (IgG1)–based TCB, cibisatamab's heterodimeric Fc region has been disabled to avoid Fc receptor engagement and confer an extended half-life. The flexible antibody structure, with 2 CEA binding domains and 1 CD3ε binding domain, enables higher CEA avidity, for selective killing of CEA-expressing tumour cells.

CEA is a cell-surface glycoprotein that reportedly plays a role in cell adhesion, invasion and metastasis of cancer cells[10] and is over-expressed on a variety of cancers, including colorectal cancer (CRC)[11,12]. CEA can be released from the plasma membrane upon enzymatic disruption of the glycosyl phosphatidylinositol binding bridge[13] and constitutes a widely used measurable quantitative biomarker[14]. Importantly, the CEA binding site for cibisatamab persists on tumour cells even if the link is cleaved and released from the plasma membrane. Indeed, to avoid toxicity from on-target off-tumour binding to circulating CEA, cibisatamab was optimized to recognize an epitope that is only present in the CEA membrane-attached form[9].

Simultaneous binding of cibisatamab to CEA and CD3ε causes T-cell activation independent of T-cell receptor specificity, leading to lymphocyte-mediated tumour cell killing, immune-stimulatory cytokine release and further release of tumour antigens. In nonclinical models, cibisatamab demonstrated the ability to increase T-cell infiltration in CEA-expressing tumours, thus converting non-inflamed programmed death-ligand 1 (PD-L1)–negative tumours into highly inflamed PD-L1–positive tumours[15]. Furthermore, combining cibisatamab with an anti–PD-L1 blocking antibody synergistically enhanced its efficacy in humanised mice[16]. In cultured human tumour organoids, redirected T-cell–mediated cytotoxicity depended on the level of CEA surface expression[17].

Because monoclonal antibodies and related constructs may induce the development of anti-drug antibodies (ADAs), researchers have studied whether pre-treatment with obinutuzumab, a gly-coengineered humanized anti-CD20 monoclonal antibody that recognizes the CD20 antigen present on B-cells, could blunt or attenuate ADA generation. In nonclinical and clinical studies, obinutuzumab induces a profound B-cell depletion, which has been shown to result in suppression of de novo antibody responses, while leaving largely intact the protective humoral memory of long-lived plasma cells[18,19].

Here we collectively describe the results of two multi-institutional Phase 1 dose-escalation and -expansion studies of cibisatamab in patients with advanced CEA-positive solid tumours: S1, studying cibisatamab as monotherapy (Study BP29541; NCT02324257[20]), and S2, studying cibisatamab combined with atezolizumab (Study WP29945; NCT02650713[21]). We evaluated the safety of cibisatamab as a single agent as well as in combination with atezolizumab in patients with MSS-CRC, demonstrating a safety profile consistent with that of each individual agent. Coupled with preliminary efficacy results, these data suggest that this treatment approach may be worth further clinical investigation.

## Results

### Patients

From 2014 to 2018, 149 and 228 patients were enrolled and treated in S1 and S2, respectively (Supplementary Figs 1 and 2). S1 included 125 patients with CRC (83.9%), 100 with confirmed microsatellite stable (MSS) disease (67.1%), 2 with confirmed microsatellite instability-high (MSI-H) (1.3%) disease and remaining patients unknown. All patients had metastatic disease at study entry, most with metastases to the liver (n = 38 [25.5%]), lung (n = 30 [20.1%]) or both (n = 73 [49.0%]). At study entry, patients' mean age was 60 years (range: 22-80 years), 61 (40.9%) were female. Eastern Cooperative Oncology Group performance status (ECOG PS) at baseline was 0 (n = 82 [55.0%]) or 1 (n = 67 [45.0%]). 147 patients (98.7%) had received at least one prior line of therapy for metastatic disease, 61 (40.9%) received prior adjuvant treatment and 90 (60.4%) received 3 or more prior lines of therapy.

S2 included 192 patients with CRC (84.2%), 187 with confirmed MSS (82%) disease, 5 with confirmed MSI-H (2.2%) disease and remaining patients unknown. All patients had metastatic disease at study entry, most with metastases to the liver (n = 56 [24.6%]), lung (n = 33 [14.5%]) or both (n = 127 [55.7%]). At study entry, patients' mean age was 57 years (range: 24-81 years), 96 (42.1%) were female. ECOG PS at baseline was 0 (n = 132 [57.9%]) or 1 (n = 96 [42.1%]). 227 patients (99.6%) had received at least one prior line of therapy for metastatic disease, 91 (39.9%) received prior adjuvant treatment and 140 (61.4%) received 3 or more prior lines of therapy. Detailed baseline characteristics for S1 and S2 are presented in Supplementary Table 1.

In S1, 149 patients were treated with flat-dose levels of cibisatamab ranging from 0.052 to 600 mg and with step-up dosing cohorts, and with obinutuzumab pretreatment. In S2, 228 patients were treated at flat-dose levels of cibisatamab ranging from 5 to 300 mg and in step-up dosing cohorts in combination with atezolizumab.

All patients were included in the S1 and S2 safety- and efficacy-evaluable populations, which were defined, respectively, as all patients who received at least one dose of cibisatamab or obinutuzumab and all patients who received at least one dose of cibisatamab or atezolizumab. In S1, 65 patients who received flat-dose cibisatamab without obinutuzumab prior to treatment and 15 patients who received cibisatamab with obinutuzumab prior to treatment were evaluable for dose-limiting toxicities (DLTs). Step-up dosing cohorts were explored to assess whether the approach of starting with low-dose cibisatamab then dose-escalating would mitigate infusion-related reactions observed during late cycles. An additional 22 patients in step-up dosing cohorts were evaluable for DLTs to determine the late cycle MTD. In S2, 73 patients in Part IA (dose escalation) were evaluable, while 153 patients in Part IB (dose schedule finding) were evaluable for DLTs to determine the late cycle MTD.

### Safety (primary objective)

In S1, 116 of the 149 patients (77.9%) received cibisatamab only, 27 patients (18.1%) received obinutuzumab pretreatment followed by

**Table 1 | Safety overview of S1 and S2**

| Type of AE | S1: cibisatamab mono-therapy (N = 149) Patients, n (%) | S2: cibisatamab + atezolizumab (N = 228) Patients, n (%) |
|---|---|---|
| Any AE | 146 (98.0) | 228 (100.0) |
| Cibisatamab-related AE | 139 (93.3) | 225 (98.7) |
| Grade 5 AE | 6 (4.0) | 6 (2.6) |
| Cibisatamab-related grade 5 AE | 3 (2.0) | 1 (0.4) |
| Grade ≥3 AE | 102 (68.5) | 154 (67.5) |
| Cibisatamab-related grade ≥3 AE | 54 (36.2) | 111 (59.7) |
| Max Grade 3 AE | 89 (59.7) | 131 (57.5) |
| Max Grade 4 AE | 7 (4.7) | 17 (7.5) |
| SAE | 98 (65.8) | 145 (63.6) |
| Cibisatamab-related SAE | 69 (46.3) | 120 (52.6) |
| AE leading to withdrawal of cibisatamab | 6 (4.0) | 15 (6.6) |
| AE leading to withdrawal of atezolizumab | NA | 13 (5.7) |

AE Adverse event, NA Not applicable, SAE Serious adverse event.
Source data can be requested from the authors for academic research purposes

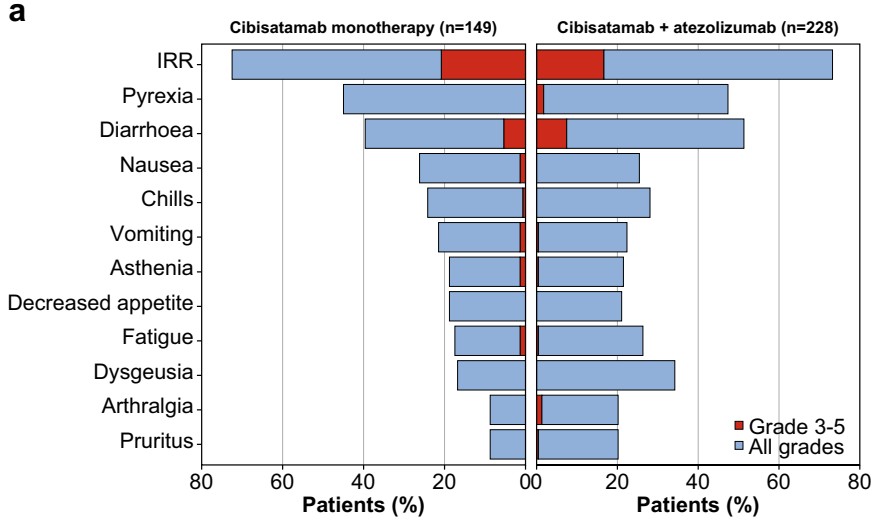

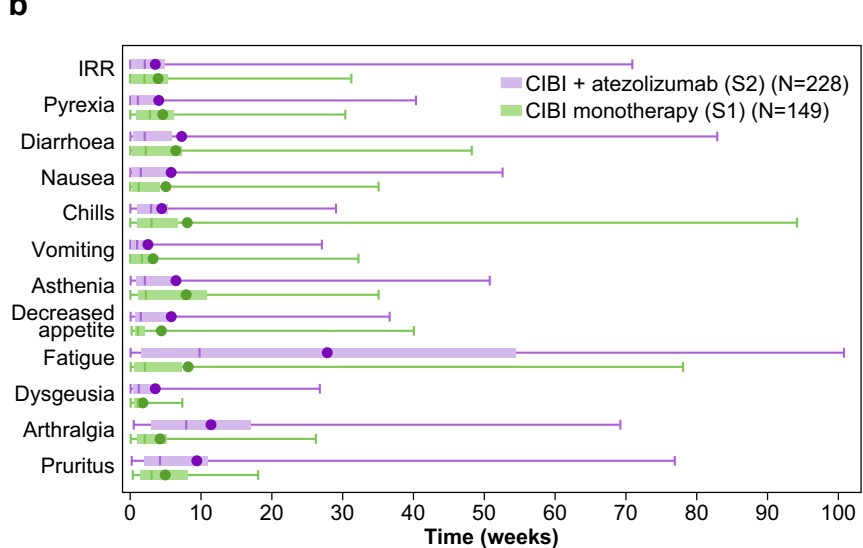

**Fig. 1 | Treatment-related adverse events. a** Most frequent treatment-related adverse events (≥ 20% in either study) in S1 and S2. **b** Median time to onset (middle line; with means as circles, quartiles as boxes, and ranges as bars) of most common treatment-related adverse events (≥20% in either study) in S1 and S2. CIBI Cibisatamab, IRR Infusion-related reaction. Source data can be requested from the authors for academic research purposes.

cibisatamab and 6 patients (4.0%) received obinutuzumab only due to treatment discontinuation prior to receiving the first dose of cibisatamab. An overview of safety in S1 and S2 is provided in Table 1. Apart from a lower incidence of both on-target/off-tumour gastro-intestinal events (diarrhoea, nausea, vomiting) and infusion-related reactions (IRRs) at doses below 60 mg in S1 and doses below 80 mg in S2, no clear trends for dose-dependent adverse events (AEs) were observed. Of note, most patients developed early onset transient tumour inflammation and systemic cytokine-related effects (see below).

In S1, 6 of 149 patients (4.0%) experienced the following grade 5 events: dyspnoea, IRR and respiratory failure (all considered related to cibisatamab); sepsis (considered related to obinutuzumab); and cardio-respiratory arrest and tumour thrombosis (not considered treatment-related by the investigator). In S2, 6 of 228 patients (2.6%) experienced grade 5 events: 1 cibisatamab-related event of hypovo-lemic shock and 5 events that were not considered treatment related by the investigator (respiratory tract infection, urinary tract infection, disseminated intravascular coagulation, bile duct obstruction and cerebrovascular accident).

## DLTs and MTD (primary objective)

In S1, the maximum tolerated dose (MTD) was defined as 400 mg for flat continuous dosing once weekly (QW) and once every 3 weeks (Q3W), based on 7 cibisatamab-related DLTs reported in 7 of the 102 DLT-evaluable patients (6.9%). Two of these 7 DLTs were grade 5 events of respiratory failure and dyspnoea. In S2, which included Part IA (dose escalation) and Part IB (dose schedule finding), a total of 17 DLTs were reported in 15 of the 226 DLT-evaluable patients (6.6%) and included a grade 5 event of hypovolemic shock. During the S2 dose-escalation phase, all 3 patients at the 300-mg dose level had serious AEs (SAEs) after their first infusion. While these SAEs were not classified as DLTs, it was decided not to escalate the dose further. Thus, the MTD for cibisatamab in combination with atezolizumab was not reached.

## Frequency and severity of AEs

Cibisatamab demonstrated a dynamic safety profile, with the onset of most toxicity beginning in cycles 1 to 3 and decreasing as treatment progressed and tolerance improved. The main exceptions to this trend were fatigue, arthralgia and pruritus; these had notably later median

**Table 2 | Reported and retrospectively assessed incidence of CRS in S1 and S2 according to National Cancer Institute Common Terminology Criteria for Adverse Events v5.0**

| Derivation method and grade | S1: cibisatamab monotherapy (N = 143)[a]<br>Patients, n (%) | S2: cibisatamab + atezolizumab (N = 228)<br>Patients, n (%) |
|---|---|---|
| CRS (as reported term) | 0 | 7 (3.1) |
| Grade 1 | 0 | 1 (0.4) |
| Grade 2 | 0 | 1 (0.4) |
| Grade 3 | 0 | 5 (2.2) |
| CRS (retrospectively assessed grade ≥2) | 28 (19.6) | 23 (10.1) |
| Grade 2 | 26 (18.2) | 20 (8.8) |
| Grade 3 | 1 (0.7) | 3 (1.3) |
| Grade 5 | 1 (0.7) | 0 |

CRS Cytokine-release syndrome.

[a]Cibisatamab with or without obinutuzumab. Excludes 6 patients who only received obinutuzumab.

Source data can be requested from the authors for academic research purposes

onset times in S2 than in S1 and are known risks of atezolizumab (Fig. 1).

AEs were predominantly associated with infusion of study treatment and the subsequent activation of immune cells, leading to local inflammation in the tumours and systemic cytokine-related effects.

In both S1 and S2, IRRs were reported in more than 70% of patients, with pyrexia, chills, vomiting, nausea and diarrhoea being the most commonly observed symptoms reported within 24 hours of the end of infusion (Fig. 1 and Supplementary Table 2). Grade 3 IRRs occurred in 31 of 143 S1 patients (21.7%) and 38 of 228 S2 patients (16.2%). The majority of IRRs and associated symptoms were short in duration and resolved the same day (see supplementary Tables 3 and 4) with protocol-recommended premedication measures and supportive care. To further mitigate the risk of IRRs and cytokine-release syndrome (CRS), corticosteroid pretreatment was given during step-up dosing (intravenous [IV] dexamethasone 10 mg or equivalent) and for patients who experienced a grade 2 IRR or CRS in the previous cycle (IV methylprednisolone 80 mg or IV dexamethasone 16 mg).

No CRS events were reported by preferred term in S1; in S2, CRS events were reported in 7 of 228 patients (3.1%). Given the overlap in signs and symptoms of IRR and CRS and the timing of the studies relative to the evolving definition and clinical management of CRS, it is clear that patients in studies S1 and S2 may have experienced CRS events that were instead reported as IRRs. To assess the incidence of CRS in these studies, a retrospective analysis was conducted to identify all events of hypoxia, hypotension or both (reported as standalone events or as reported symptoms of IRR) that occurred within 48 h of cibisatamab infusion. Patient records that indicated treatment of these events with supplemental oxygen or vasopressors and laboratory findings such as cytokine levels and ADA status were used to support the analysis.

Twenty-eight of 143 patients (19.6%) in S1 and 23 of 228 patients (10.1%) in S2 would have been captured as grade ≥ 2 CRS according to National Cancer Institute Common Terminology Criteria for Adverse Events (NCI CTCAE) v5.0 (in addition to the 7 patients in S2 [3.1%] for whom CRS was reported as the preferred term) (Table 2).

### Tumour inflammation

In addition to systemic cytokine-driven toxicities, engagement of the tumour target and activation of T cells may drive a rapid expansion and infiltration of immune cells, resulting in tumour inflammation or flair, pain at the tumour site and, depending on the location of the inflamed tumour, mass effects that can impact organ function.

This phenomenon of temporary tumour enlargement was consistently observed in patients who received computed tomography (CT) scans within 48 to 96 h after drug administration and was attributed to inflammation of the tumour tissue (Fig. 2). In most cases, events of tumour inflammation, tumour pain and associated sequelae resolved within 2 to 3 days of administration, following management with corticosteroids and supportive care. However, severe pulmonary toxicity was observed in a small number of patients with multiple bilateral lung lesions and in patients with tumours sited near critical organ structures. Of note, one patient had respiratory failure and died following their first dose of cibisatamab 600 mg, as the result of an extrinsic tracheal/bronchial obstruction by an enlarged tumour mass. For this reason, patients with important thoracic involvement at critical sites and high lung tumour burden were excluded from ensuing trials.

### Incidence of AEs in ADA-positive and -negative patients

Cibisatamab induced the formation of ADAs in 50% to 70% of patients. Figure 3 shows the combined incidence from S1 and S2 of selected AEs and associated symptoms related to IRR and CRS in ADA-positive and ADA-negative patients by infusion (first 7 infusions shown).

In both studies, IRRs and associated symptoms were more frequent and of higher severity in ADA-positive patients than in ADA-negative patients (Fig. 3). IRRs of grade 3 or higher were also more persistent in later cycles in ADA-positive patients than in ADA-negative patients. Irrespective of a patient's ADA status, most IRRs were observed following the first infusion (median time to ADA onset in S1 was 22 days; in S2, 16 days).

### Pharmacokinetics and Immunogenicity (primary objective in S1; secondary objective in S2)

A two-compartment pharmacokinetic (PK) model with first-order linear elimination well described the cibisatamab serum concentration time course in ADA-negative patients. In S1 and S2, the median clearance values were 0.063 L/h and 0.048 L/h and the volume of distribution at steady state values were 8.25 L and 8.31 L, respectively.

Clearance did not depend on dose and exposure and was similar after administration of cibisatamab alone or in combination with atezolizumab. After 100 mg Q3W, the maximum concentration and area under the time-concentration curve of the first dosing interval was 26.7 μg/mL (13.8–47.9) and 756 μg×h/mL (266–2398), respectively (median, 5th-95th percentile, n = 20). In ADA-negative patients, pharmacokinetics were time-independent. The cibisatamab clearance is approximately 7-fold to 8-fold higher than a typical IgG1 antibody in humans[22].

The incidence of ADA development was high (50% in S1 in patients not pretreated with obinutuzumab; 71% in S2; Table 3). The drug tolerance of the ADA assay used in S2 was significantly better than that of the assay used in S1; therefore, the incidence of ADAs is not directly comparable. The median time to ADA onset was 22 days (range, 6–146 days) and 16 days (range, 7–260 days) in studies S1 and S2, respectively. Higher ADA titers were associated with a larger impact on loss of cibisatamab exposure, suggesting that ADAs are neutralizing, i.e., directed to epitopes in the CEA- or the CD3-binding regions of cibisatamab, or both, or increasing drug clearance from plasma.

After obinutuzumab pretreatment, 11 of 26 patients (42%) were ADA positive. Nine of 11 patients (82%) had transient ADAs, and in the remaining 2 patients later time points were unavailable (the last available ADA sample was early—at 3 and 9 weeks after the first dose of cibisatamab). The maximum observed ADA titer in obinutuzumab-pretreated patients was 1:270, while ADA titers were in general higher in patients who were not pretreated with obinutuzumab, with a maximal observed titer of 1:196,830 (study S1). In the obinutuzumab-

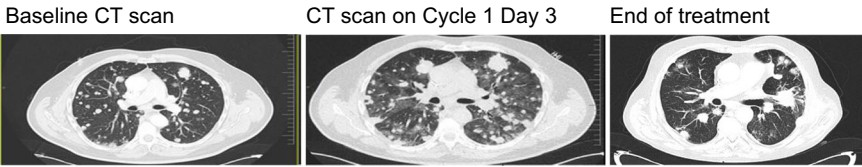

**Fig. 2 | Patient with bilateral lung lesions at baseline and lung lesion inflammation and localised perilesional oedema on Cycle 1 Day 3.** The patient received 60 mg cibisatamab on Cycle 1 Day 1 and experienced transient hypoxia and transient dyspnoea 48 h after infusion. CT Computed tomography.

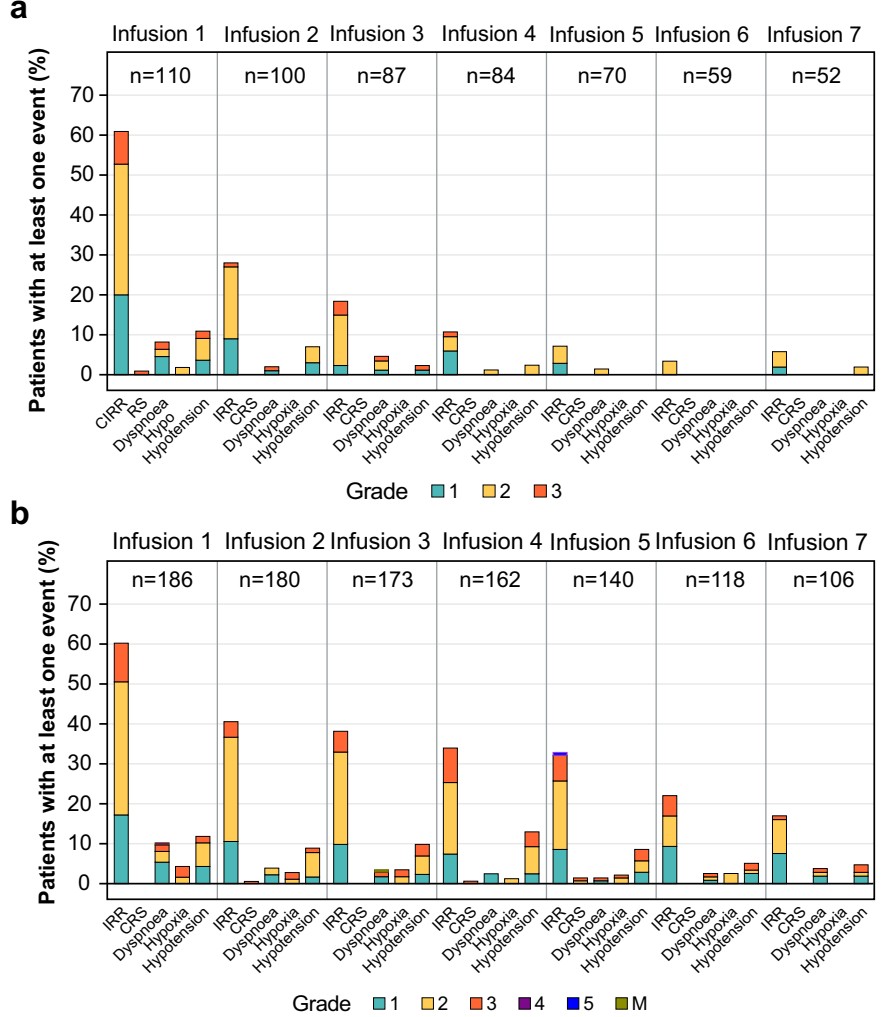

**Fig. 3 | Combined incidence and severity of IRR- and CRS-related adverse events and associated symptoms.** Data for (**a**) ADA-negative and (**b**) ADA-positive patients are shown by infusion for patients receiving 40–600 mg of cibisatamab (flat-dose and step-up dosing cohorts) in S1 and S2. ADA, Anti-drug antibody; CRS, Cytokine-release syndrome; IRR, Infusion-related reaction; M Grade information not provided by site. Source data can be requested from the authors for academic research purposes.

pretreated population, cibisatamab exposure was sustained in ADA-positive patients, and the cibisatamab concentration levels were similar to those in ADA-negative patients (Supplementary Fig. 3).

In ADA-negative patients, PK was time-independent, i.e., serum exposure was maintained after multiple doses. In persistent ADA-positive patients, however, ADA-mediated, time-dependent PK was observed with reduced or no detectable exposure at end of infusion. Median time to onset of complete loss of exposure (exposure not detectable at end of infusion) was 35 days (range, 21–111 days) and 63 days (range, 21–189 days) in studies S1 and S2, respectively. Step-up dosing to high doses could not overcome the negative impact of ADAs on active cibisatamab exposure.

## Efficacy (secondary objectives)

In S1, all patients ($n = 149$) had measurable disease at baseline and were evaluable for efficacy. Six patients (4.0%, 90% CI: 1.8, 7.8) achieved a confirmed partial response (PR), including 3 of 46 patients (6.5%) enrolled in the step-up cohorts, 2 of 27 patients (7.4%) enrolled in the cohorts with obinutuzumab pretreatment and 1 of 65 patients (1.5%) enrolled in the flat dose cohorts. Five of 6 PRs occurred in patients with MSS-CRC and one in a patient with pancreatic cancer. The median duration of response was 6.5 months (90% CI: 3.9, 7.4). Forty patients (26.8%) had a best overall response of stable disease (SD), 74 patients (49.7%) had a best overall response of progressive disease (PD) and 29 patients (19.5%) had missing or non-evaluable best overall responses

**Table 3 | Incidence of patients with cibisatamab ADAs and incidence of patients with complete loss of cibisatamab exposure in S1 and S2**

|  | S1 (without obinutuzumab) | | S1 (with obinutuzumab pretreatment) | | S2 (without Obinutuzumab) | |
|---|---|---|---|---|---|---|
|  | *N* | *n* (%) | *N* | *n* (%) | *N* | *n* (%) |
| ADA positive at baseline | 111 | 3 (2.7) | 26 | 0 | 218 | 8 (3.7) |
| ADA-negative, persistent | 110 | 55 (50) | 26 | 15 (58) | 219 | 64 (29) |
| ADA-positive, treatment-enhanced | 110 | 1 (1) | 26 | 0 | 219 | 2 (1) |
| ADA-positive, transient | 110 | 10 (9) | 26 | 9 (35) | 219 | 39 (18) |
| ADA-positive, persistent | 110 | 44 (40) | 26 | 2 (8) | 219 | 114 (52) |
| Complete LOE | 116 | 30 (26) | 27 | 0 | 228 | 42 (18) |

*ADA* Anti-drug antibody, *LOE* Loss of exposure.
Treatment-emergent ADAs were classified and subclassified as either:
1) Treatment-induced ADA: Patient has negative or missing baseline ADA result(s) and ≥1 positive post-baseline ADA result.
(a) Persistent: Patient has post-treatment ADA-positive samples for ≥16 weeks or the last ADA time point is positive
(b) Transient: Patient has ≥1 post-treatment ADA-positive sample AND has only 1 ADA-positive sample or the time between the first and last ADA-positive sample is <16 weeks AND the last ADA sample is negative
2) Treatment-enhanced ADA: Patient has positive ADA result at baseline and ≥1 post-baseline titer result that is ≥0.60 titer units greater than the baseline titer result
Complete LOE is defined as a cibisatamab exposure with target-binding competent PK assay (i.e., active concentration) that is not detectable at end of infusion. Note that N for on-treatment ADA status can be larger than for baseline status as patients are still eligible for some categories with missing baseline samples. Source data can be requested from the authors for academic research purposes

across all dose cohorts. Response results for the cibisatamab monotherapy cohorts are summarized in Supplementary Tables 5 and 6.

In S2, all enrolled patients ($n$ = 228) had measurable disease at baseline and were evaluable for efficacy. Across all dose levels, schedules and tumour types, the investigator-assessed overall response rate (ORR) was 6.6% (90% CI: 4.1, 9.9); 34.6% of patients (79/228) achieved a SD and 47.8% (109/228) showed PD as best overall response and a disease control rate (DCR) of 41.2% (90% CI: 35.8, 46.9). Eleven percent of the patients were non-evaluable for response.

In Part IA ($n$ = 75), the DCR was 27.8% (5/18) in the patients who were enrolled in the dose-escalation cohorts ranging from 5 to 40 mg. No objective response was observed. In the dose-escalation cohorts that ranged from 80 to 160 mg, the DCR was 61.5% (8/13) and one patient had a PR (7.7%, 90% CI: 0.4, 31.6). Based on these early efficacy findings, the 160-mg cohort was expanded to recruit an additional 41 patients. In addition, Part IB was opened to test the randomized flat dose with 100 mg QW vs 100 mg Q3W and to include several step-up cohorts (see the CONSORT diagrams in Supplementary Figs. 1, 2).

Table 4 summarizes the efficacy results in the flat-dose 100- to 160-mg MSS-CRC cohorts, the step-up dose MSS-CRC cohorts and all enrolled patients with MSS-CRC. Figure 4 shows the spider plot of change in target lesion of the patients with MSS-CRR who were treated in the randomized 100-mg cohorts. Efficacy based on objective response appeared to be similar in patients with MSS-CRC who were treated with flat doses of cibisatamab 100 mg QW, 100 mg Q3W or 160 mg QW in combination with atezolizumab, and efficacy seemed to be greater in these patients receiving flat doses than in those who received step-up dosing.

Patients with several other diagnoses received cibisatamab in combination with atezolizumab, including patients with pancreatic ($n$ = 17), gastric ($n$ = 12), non-small cell lung ($n$ = 3), breast ($n$ = 2) and bile duct ($n$ = 2) cancer. The efficacy results are summarised in Supplementary Table 7.

Of the enrolled patients in S2, 127 had CEA-related cell adhesion molecule 5 (*CEACAM5*) messenger RNA expression data available; 105 of these patients had MSS-CRC and 45 were in the 100- to 160-mg dose cohorts (Fig. 5A). Thirteen patients were classified as *CEACAM5*-high and 32 patients were *CEACAM5*-low, according to a normalised RNA-seq cutoff of 1500 reads per kilobase of transcript per million reads mapped (RPKM). Importantly, all 4 of the tested patients who had a confirmed PR were *CEACAM5*-high.

All key analyses were repeated disaggregated by sex with selected results presented in Supplementary Table 8. No substantial differences between the sexes were found.

## Exploratory biomarker results

The immunohistochemistry (IHC) data from the 100- to 160-mg QW or Q3W cohorts show that in the majority of patients (80%, $n$ = 31), cibisatamab in combination with atezolizumab induces an increase in intra-tumoural T-cell infiltration (comparing the C2D1 and C3D1 on-treatment time points to baseline levels). On-treatment *PDCD1* levels (*PDCD1* encodes programmed death ligand-1 [PD-L1]) were similarly higher than baseline levels (Fig. 5B–E presents the CD8 and PD-1 IHC images and data that were quantified). Furthermore, cibisatamab treatment frequently triggered the relocalization of the CD8 + T cells from the stroma to the tumour beds, resulting in the conversion of immune-excluded tumours (in which the CD8 T cells are primarily in the tumour stroma) to inflamed tumours (in which the CD8 T cells are infiltrating the tumour nests) (Fig. 5F). In this limited data set, these biomarkers only reflect pharmacodynamic changes and are not predictive of response.

Cibisatamab is designed to allow for activation of T cells regardless of their antigen specificity. This was confirmed by RNAseq analysis on tissue collected at baseline and on treatment ($n$ = 23 in the cibisatamab-atezolizumab 100-mg cohorts). As expected, T effector (Teff) gene expression (*CD8A, GZMA, GZMB, IFNG, EOMES, PRF1, CXCL9, CXCL10, TBX21*) increased on treatment (supporting the CD8 and PD-L1 data). However, no changes in antigen presentation genes (*TAPBP, TAP1, TAP2, PSMB8, PSMB9*) were seen, supporting the theorized synthetic immunity mechanism of action of TCBs, whereby conceivably no antigen-specific T cells are necessary for activity (Fig. 6). Interestingly, it does seem that in the Teff gene signature, the change was greater in the patients with PR and SD than in those with PD (mean changes: patients with PR, 1.5; SD, 1.3; PD, 0.7). However, this trend needs further validation. More minimal changes in peripheral CD8 and CD4 cells were seen by response in patients receiving cibisatamab (Supplementary Fig. 4).

## Dose selection

Multiple dose levels and schedules for cibisatamab monotherapy (with or without obinutuzumab pretreatment) and cibisatamab in combination with atezolizumab have been tested in studies S1 and S2 in approaches intended to improve safety and efficacy. The flat-dosing

**Table 4 | Efficacy for all patients, all patients with MSS-CRC, all patients with MSS-CRC who received flat doses and step-up dose MSS-CRC treated with cibisatamab in combination with atezolizumab**

| | All (*n* = 228) | All MSS-CRC (*n* = 187) | Flat-dose MSS-CRC 100 QW (*n* = 20) | Flat-dose MSS-CRC 100 Q3W (*n* = 19) | Flat-dose MSS-CRC 160 QW (*n* = 35) | All step-up B1 + C2[a] MSS-CRC (*n* = 50) | Step-up C1[b] MSS-CRC (*n* = 39) |
|---|---|---|---|---|---|---|---|
| ORR, *n* (%) 90% CI | 15 (6.6) [4.1, 9.9] | 13 (7.0) [4.2, 10.8] | 3 (15.0) [4.2, 34.4] | 3 (15.8) [4.4, 35.9] | 5 (14.3) [5.8, 27.7] | 0 [0.0, 5.8] | 2 (5.1) [0.9, 15.3] |
| CR, *n* (%) 90% CI | 1 (0.4) [0.0, 2.1] | 0 [0.0, 1.6] | 0 [0.0, 13.9] | 0 [0.0, 14.6] | 0 [0.0, 8.2] | 0 [0.0, 5.8] | 0 [0.0, 7.4] |
| PR, *n* (%) 90% CI | 14 (6.1) [3.8, 9.4] | 13 (7.0) [4.2, 10.8] | 3 (15.0) [4.2, 34.4] | 3 (15.8) [4.4, 35.9] | 5 (14.3) [5.8, 2.7] | 0 [0.0, 5.8] | 2 (5.1) [0.9, 15.3] |
| SD, *n* (%) 90% CI | 79 (34.6) [29.4, 40.2] | 68 (36.4) [30.5, 42.6] | 8 (40.0) [21.7, 60.6] | 7 (36.8) [18.8, 58.2] | 13 (37.1) [23.6, 52.4] | 21 (42.0) [30.1, 54.6] | 9 (23.1) [12.6, 36.8] |
| PD, *n* (%) 90% CI | 109 (47.8) [42.2, 53.5] | 88 (47.1) [40.9, 53.3] | 9 (45.0) [25.9, 65.3] | 8 (42.1) [23.0, 63.2] | 13 (37.1) [23.6, 52.4] | 22 (44.0) [32.0, 56.6] | 25 (64.1) [49.7, 76.8] |
| NE, *n* (%) | 25 (11.0) | 18 (9.6) | 0 | 1 (5.3) | 4 (11.4) | 7 (14.0) | 3 (7.7) |
| DCR, *n* (%) 90% CI | 94 (41.2) [35.8, 46.9] | 81 (43.3) [37.2, 49.6] | 11 (55.0) [34.7, 74.1] | 10 (52.6) [32.0, 72.6] | 18 (51.4) [36.5, 66.2] | 21 (42.0) [30.1, 54.6] | 11 (28.2) [16.7, 42.3] |
| DOR range, mo | 1.9–35.6 | 1.9–28.6 | 3.8–6.9 | 3.6–27.7 | 1.9–28.6 | NA | 6.0–17.6 |

Patients were randomly assigned to the 100-mg QW or 100-mg Q3W cohorts.

*CR* Complete response, *DCR* Disease control rate, *DOR* Duration of response, *MSS-CRC* Microsatellite-stable colorectal cancer, *NA* Not applicable, *NE* Unevaluable or missing, *PD* Progressive disease, *PR* Partial response, *Q3W* Once every 3 weeks, *QW* Once weekly, *SD* Stable disease, *ORR* Overall response.

[a]The maximum planned cibisatamab dose was 1200 mg in cohort B1 and 600 mg in cohort C2.

[b]The maximum planned cibisatamab dose was 150 mg in cohort C1.

Source data can be requested from the authors for academic research purposes.

regimens of cibisatamab (100 mg QW, 100 mg Q3W, and 160 mg QW) in combination with atezolizumab seemed to have a more favorable benefit-risk profile than step-up dosing regimens that started at 40 mg and escalated to the 1200-mg dose. In S2, the safety profile and clinical efficacy of cibisatamab appeared to be similar with both the randomized 100-mg dose administered QW and Q3W and the 160-mg dose administered QW in S2.

In a longitudinal mixed-effect nonlinear model that included patients with MSS-CRC who were enrolled in S2 and had been exposed to flat doses, doses of ≥ 80 mg cibisatamab led to a higher tumour shrinkage rate and to a trend toward lower growth rates.

Owing to the high affinity of cibisatamab to CEA and consistent with cibisatamab positron emission tomography (PET) imaging studies in tumour-bearing mice[23] and a clinical imaging study with a labeled molecule ($^{89}$Zr CEA-IL2v) that binds to the same CEA epitope as cibisatamab[24], tumour retention is predicted to be considerably longer than that based on the serum PK half-life, which further supported the Q3W schedule. In addition, the 100-mg Q3W schedule will be more convenient for patients and will allow for a longer recovery period between cibisatamab administrations. Thus, 100 mg Q3W cibisatamab in combination with 1200 mg Q3W atezolizumab has been selected as the recommended Phase 2 dose (RP2D) and schedule.

## Discussion

There remains a significant unmet clinical need to optimize immunotherapeutic strategies for patients with cancer, in particular MSS-CRC. T-cell redirecting therapies using bispecific antibodies present a unique opportunity to engage T cells and induce potent anti-tumour immunity. Cibisatamab is the first T-cell bispecific antibody with a 2-to-1 format, optimized for safety and efficacy, directly binding CEA on tumour cells and CD3 on T cells, regardless of antigen specificity, resulting in increased T-cell infiltration, T-cell activation and tumour-cell killing[15].

The S1 and S2 studies were performed in order to evaluate the safety, optimal dose and schedule, and preliminary efficacy of cibisatamab alone and in combination with atezolizumab. The safety profile of cibisatamab was dominated by IRRs, diarrhoea and tumour inflammatory events. Grade 3 and 4 toxicities were mostly IRRs,

diarrhoea, transaminitis and anaemia. We observed that most IRRs and CRSs typically occurred following the first cibisatamab infusion and likely mediated by on-tumour/on-target effects. Later-cycle IRR and CRSs were more pronounced in ADA-positive patients, potentially due to ADA-mediated crosslinking of cibisatamab-bound CD3 leading to peripheral T-cell activation. The tumour inflammation and flare events occurred early and may be mediated by the mechanism of action of cibisatamab at the tumour, leading to an influx of inflammatory cells, thereby indicating an on-tumour/on-target effect. In contrast, the observed gastrointestinal toxicity likely represents an off-tumour/on-target effect of cibisatamab, targeting normal colonic mucosa, which is known to express high levels of *CEACAM5*.

The MTD for cibisatamab monotherapy was determined as a flat dose of 400 mg QW and Q3W. Safety of cibisatamab in combination with atezolizumab was consistent with the known safety profiles of the individual drugs, with no further signals reported. The MTD for cibisatamab in combination with atezolizumab was not formally determined since during the dose-escalation phase, all 3 patients at the 300-mg dose level had a serious AE after the first infusion (chills, hypoxia, diarrhoea) and it was decided not to escalate the dose further. Cibisatamab exposure was similar after administration as a single agent or in combination with atezolizumab. However, administration of cibisatamab was frequently associated with a high incidence of ADAs, leading to reduced exposure or to exposure below the limit of quantification in a significant number of patients. Cibisatamab exposure was sustained in obinutuzumab-pretreated patients, with PK profiles similar to those in ADA-negative patients, demonstrating the usefulness of B-cell depletion to reduce the formation of ADAs.

Preliminary data of anti-tumour activity with cibisatamab monotherapy and in combination with atezolizumab indicate these treatments resulted in clinical efficacy, with a few confirmed responses in patients with CEA-positive tumours across cohorts. Efficacy based on objective response appeared to be best in patients with MSS-CRC who were treated with flat doses of cibisatamab 100 mg QW, 100 mg Q3W or 160 mg QW in combination with atezolizumab. Additionally, efficacy seemed to be further enhanced in patients with high *CEACAM5* expression. However, small sample sizes must be acknowledged as a

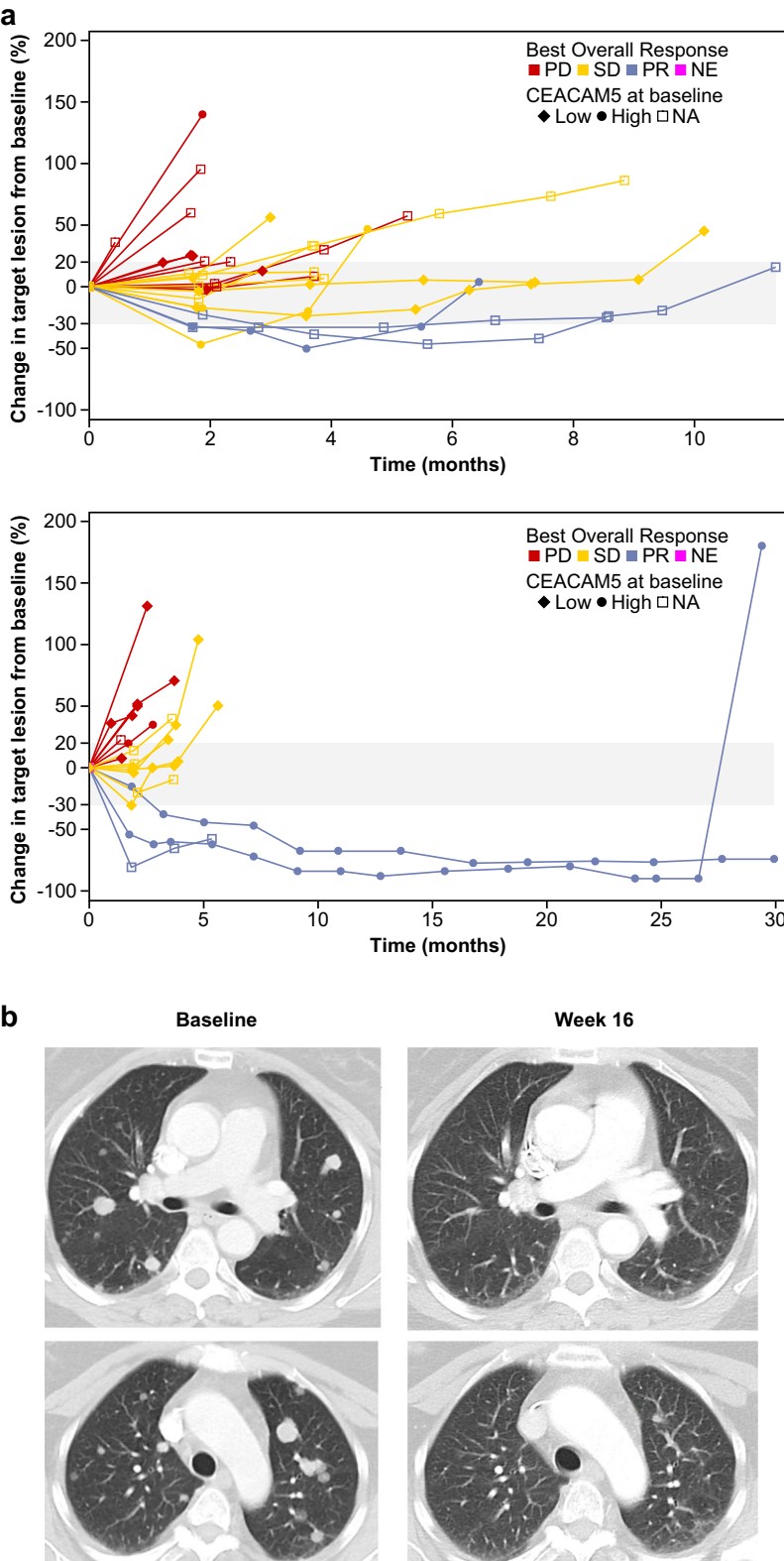

**Fig. 4 | Patterns of response. a** Spider plot of change in sum of target lesions from baseline according to Response Evaluation Criteria in Solid Tumors version 1.1 in patients with MSS-CRC receiving cibisatamab flat dose 100 mg once weekly (top, *n* = 20), and 100 mg once every 3 weeks (bottom, *n* = 19) plus atezolizumab in S2. **b** Computerised tomography scan images from a RECIST-confirmed partial responder with MSS-CRC taken at baseline and week 16 after receiving cibisatamab 160 mg QW and atezolizumab 1200 mg Q3W. CEACAM5, Carcinoembryonic antigen-related cell adhesion molecule 5; NA Not applicable, NE Not evaluable, PD Progressive disease, PR Partial response, SD Stable disease. Source data can be requested from the authors for academic research purposes.

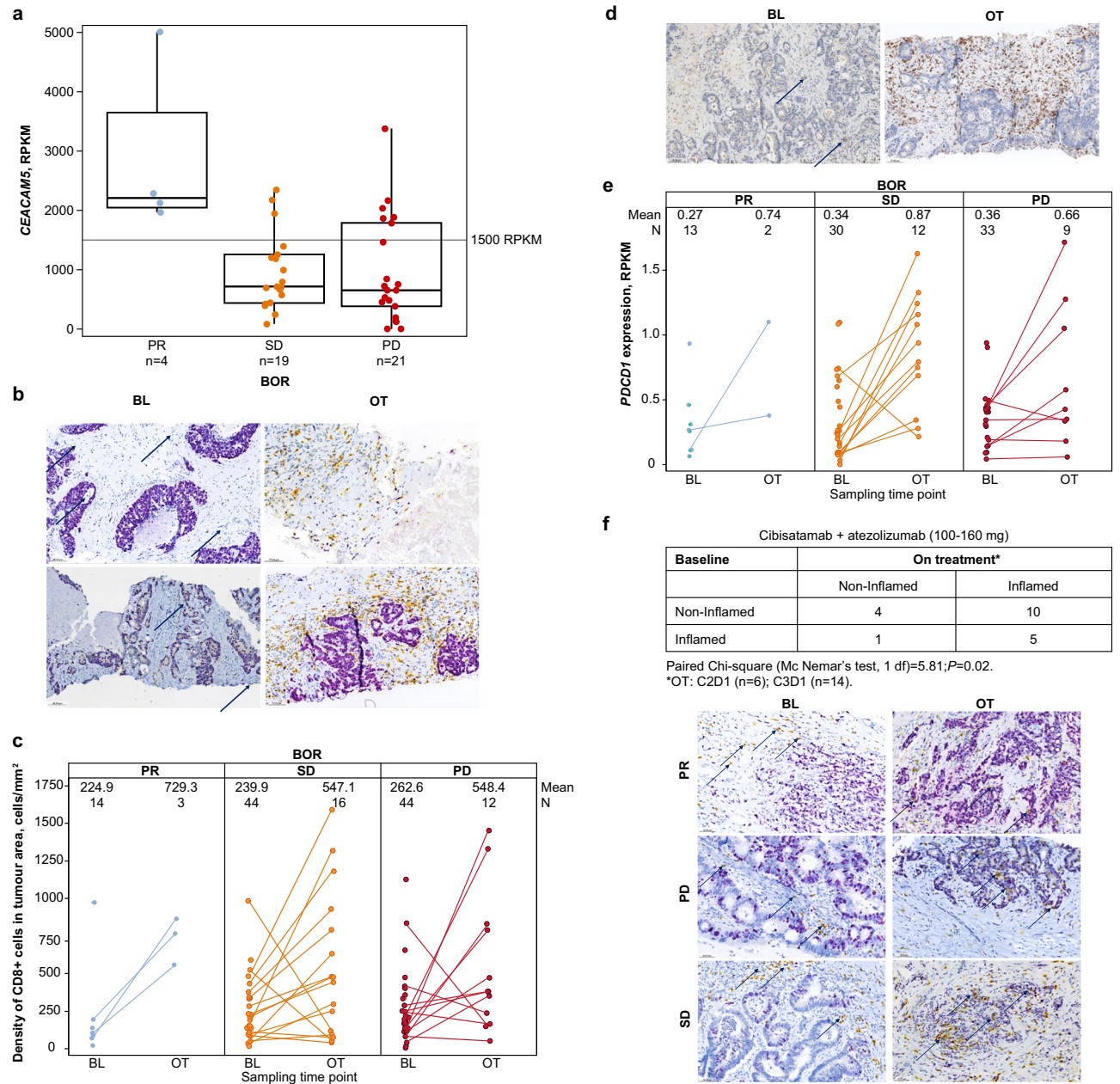

**Fig. 5 | CEACAM5 levels and intratumoural changes in CD8 + T cells and _PDCD1_ gene expression. a** _CEACAM5_ level vs best overall response according to Response Evaluation Criteria in Solid Tumors version 1.1 in the 100–160 mg cibisatamab cohort (MSS-CRC population) including all patients who received at least one dose of any study treatment (medians shown as middle lines, quartiles as boxes, and ranges as bars). **b–f** Refer to changes between baseline and on-treatment samples from patients with MSS-CRC in S2 who received doses ≥ 100 mg cibisatamab, by best overall response. **b** Dual chromogenic IHC for CD8 (yellow) and Ki67 (purple nuclear signal, outlining most tumour cells) for a patient with partial response (upper panel) and a patient with stable disease (lower panel). Scale bar represents 88.5 μm (top left), 77.3 μm (top right), 84.4 μm (bottom left) and 73.0 μm (bottom right). **c** Changes in CD8 + T cells between baseline and on-treatment tissue were quantified with a KI57/CD8 duplex assay. d Single chromogenic IHC for PD-1 at baseline (left) and on-treatment (right). Scale bar represents 76.7 μm (left) and

77.4 μm (right). **e** Changes in _PDCD1_ gene expression between baseline and on-treatment tissue were quantified by RNAseq. **f** Cibisatamab induces tissue pharmacodynamic changes. IHC for CD8 (yellow) and Ki67 (purple, outlining most tumour cells). The immune-excluded immunophenotype is shown on the left (baseline sample) and the immune-inflamed immunophenotype is shown on the right (on-treatment sample). Scale bar represents 65.4 μm (top left), 60.2 μm (top right), 64.4 μm (middle left), 52.8 μm (middle right), 64.5 μm (bottom left), and 61.0 μm (bottom right). In panels **a, c, e**, blue colour refers to PR, orange to SD, and red to PD. The N or n refers to individual patients. In subplots (**c, e**) dots for the same patient are connected by a line. The exploratory test in subplot f is two-sided. Atezo Atezolizumab; BL Baseline tissue; BOR Best overall response; _CEACAM5_ Carcinoembryonic antigen-related cell adhesion molecule 5; cibi Cibisatamab OT On-treatment tissue; PD Progressive disease; PR Partial response; RPKM Reads per kilobase of transcript per million reads mapped; SD Stable disease.

limitation of these results, and the studies were not designed to evaluate efficacy.

The biomarker data show that cibisatamab treatment triggered the relocalization of CD8 T cells from the periphery and/or the expansion of pre-existing CD8 T cells within the tumour, as well as

increased _PDCD1_ levels, leading to an inflamed tumour phenotype. The biomarker data also support the mechanisms of action of cibisatamab, in that the drug can elicit anti-tumour responses by cross-linking T cells to tumour cells and mediate polyclonal T-cell expansion that is independent of T-cell receptor specificity; evidenced by an increase in

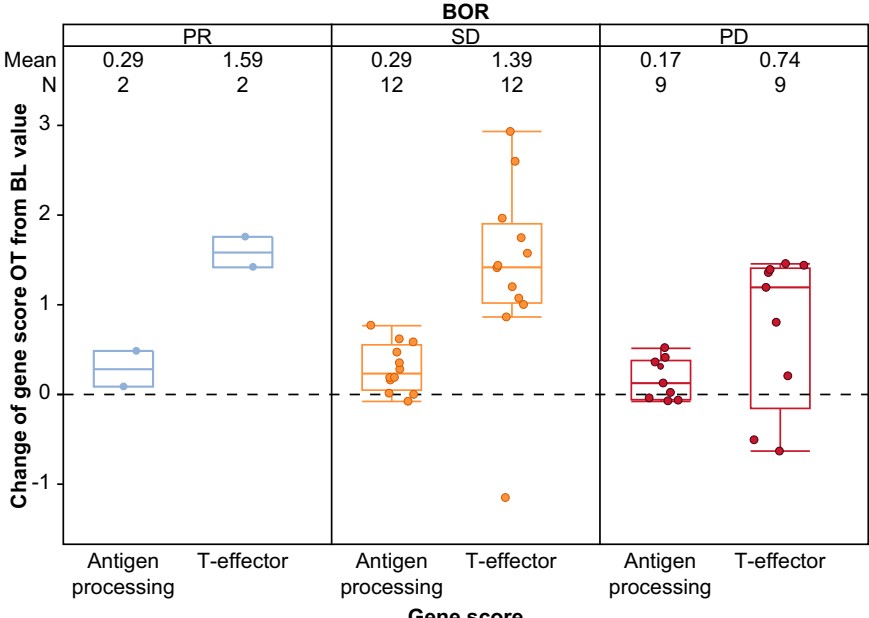

**Fig. 6 | Pharmacodynamic changes in gene expression signatures between baseline and on-treatment tissue.** These findings demonstrate increased T-effector gene signatures, with no changes in antigen presentation genes, and support the theorised synthetic immunity mechanism of action of cibisatamab targeting tumour-associated CEA, without the requirement for antigen-specific T cells and acquired immunity for activity. Medians are shown as middle lines, quartiles as boxes, and ranges as bars. N refers to individual patients with the same patients shown for antigen-processing and T-effector. Blue colour refers to PR, orange to SD, and red to PD. Antigen-processing gene set: *TAPBP, TAP1, TAP2, PSMB8, PSMB9*. T-effector gene set: *CD8A, GZMA, GZMB, IFNG, EOMES, PRF1, CXCL9, CXCL10, TBX21*. BL Baseline tissue; OT On-treatment tissue; PD Progressive disease; PR Partial response; SD Stable disease.

activated T cells in on-treatment tumour samples but no change in antigen presentation machinery. However, these pharmacodynamic changes did not correlate with response.

Cibisatamab is the first T-cell bispecific antibody consisting of a 2-to-1 format, optimised for safety and efficacy, directly binding tumour cells via CEA and T cells via CD3; resulting in increased T-cell infiltration, T-cell activation and tumour-cell killing. The preliminary efficacy observed with cibisatamab in combination with atezolizumab in patients with MSS-CRC warrants further exploration. Study CO40939 is an ongoing Phase 1b study that enrolled patients with advanced MSS-CRC with high *CEACAM5* expression treated with cibisatamab at the RP2D in combination with atezolizumab after pretreatment with obinutuzumab (NCT03866239). Furthermore, a study of cibisatamab combined with a similar bispecific construct targeting CD137/4-1BB and fibroblast activation protein α is ongoing (NCT04826003).

## Methods

### Study design and ethics
S1 (BP29541, first-in-human study) was an open-label, multi-center, multi-cohort dose-escalation Phase 1 study designed to evaluate the safety, pharmacokinetics and therapeutic activity of cibisatamab (CEA-TCB) as a single agent, with or without obinutuzumab pretreatment. S2 (WP29945) was an open-label, multi-center, multi-cohort dose-escalation and dose schedule–finding Phase 1b clinical study of cibisatamab in combination with atezolizumab. Both studies were designed and conducted in compliance with the principles of the Declaration of Helsinki and the Good Clinical Practice guidelines of the International Council for Harmonisation, and the study design and conduct complied with all relevant regulations regarding the use of human study participants. Written informed consent was obtained from each patient before the initiation of study procedures. Both study protocols (available in Supplementary Information) were approved by the institutional review boards or independent ethics committee at each study site. For study S1, the initial protocol was first approved by clinical

research ethics committee at Vall d'Hebron Hospital (Barcelona, Spain) on December 12, 2014, for study conduct at Vall d'Hebron University Hospital (Barcelona, Spain) and the University of Navarra Hospital (Navarra, Spain). For study S2, the clinical research ethics committee of the Government of Navarra Department of Health (Navarra, Spain) was first to approve the initial protocol on December 15, 2015, for study conduct at the same sites as for S1. Additional study sites in North America and Europe were opened thereafter. S1 and S2 were first authorised by The Spanish Agency for Medicine and Health Products on December 18, 2014, and December 28, 2015, respectively, with additional health authority authorizations occurring through 2017. The S1 study record was first submitted to ClinicalTrials.gov on December 12, 2014 (https://clinicaltrials.gov/study/NCT02324257); patients in S1 were enrolled between December 2014 and May 2018. The S2 study record was first submitted to ClinicalTrials.gov on January 6, 2016 (https://clinicaltrials.gov/study/NCT02650713); patients in S2 were enrolled between January 2016 and May 2018.

### Patients
In both studies, eligible patients aged 18 years or older were enrolled from 17 (S1) and 23 (S2) academic hospitals in 6 (S1) and 7 (S2) countries between 2014 and 2018 with sponsorship by F. Hoffmann-La Roche Ltd. Patients were required to have locally advanced or metastatic solid tumours after progression on a standard therapy, be intolerant of and/or non-amenable to the standard-of-care therapies and have at least one tumour lesion of accessible non-critical location to biopsy, adequate organ function, radiologically measurable and clinically evaluable disease (per Response Evaluation Criteria in Solid Tumors version 1.1 [RECIST 1.1]), and an ECOG PS 0 or 1. Patients with select non-CRC tumours had tumour CEA expression (≥ 20% of tumour cells expressing moderate or high intensity CEA expression−IHC2+ and IHC3 + ) assessed locally in Europe and centrally in North America. Patients were excluded if they had ongoing or recent autoimmune disease, had undergone allogeneic bone marrow or solid-organ

transplantation, or had concurrent cancer. A full list of inclusion and exclusion criteria is provided in the Supplementary Methods.

## Study drug administration and dose escalation

Because the human CEA binder of cibisatamab does not cross-react with cynomolgus monkey CEA and CEA is absent in rodents, the minimal anticipated biological effect level (MABEL) approach was used to define the starting dose for the first-in-human study S1. Results from in vitro studies on human cells using the most sensitive test systems and assay conditions yielded a MABEL dose for cibisatamab of 0.052 mg, which was the starting dose for this study[25].

In S1, patients were enrolled in single-patient ascending cohorts with flat-dose levels (i.e., same dose at each infusion) starting from 0.052 mg and up to 2.5 mg cibisatamab QW. In Part II of S1, patients were enrolled in multiple patient cohorts with flat-dose levels starting from 2.5 to 600 mg cibisatamab administered QW or Q3W (with or without obinutuzumab pretreatment) or step-up dosing regimen with QW and Q3W schedules (first dose of 40 mg up to 1200 mg, if tolerated). As preliminary data suggested that cibisatamab ADAs decreased the cibisatamab serum exposure (see Pharmacokinetics and Immunogenicity), new patient cohorts were created to include obinutuzumab pretreatment, to assess whether obinutuzumab could attenuate the development of ADAs. Step-up dosing cohorts were also added, to assess whether, by increasing the cibisatamab dose, the negative impact of cibisatamab ADAs on exposure could be overcome and thereby optimize efficacy. Obinutuzumab was administered as pretreatment either on Day –13 at a dose of 2000 mg or on two consecutive days, Day –13 and Day –12, at a dose of 1000 mg prior to treatment with cibisatamab on Cycle 1 Day 1.

In S2, patients were enrolled first into Part IA (dose-escalation), in which they received flat-dose cibisatamab between 5 and 300 mg QW, and then into Part IB (dose- and schedule-finding), in which patients with MSS-CRC were randomized to cibisatamab 100 mg QW or 100 mg Q3W (Cohort A). Several step-up cohorts were added with a cibisatamab starting dose of 40 mg and stepwise dose increase QW during the first 3 administrations and Q3W thereafter (Cohorts C1 and C2: patients with MSS-CRC who received a maximum cibisatamab dose of 150 mg and 600 mg, respectively; Cohort B1: patients MSS or MSI-high CRC who received a maximum cibisatamab dose of 1200 mg) The study also included step-up safety cohorts enrolling patients with indications other than CRC, including breast, gastric and pancreatic cancer, who received a maximum cibisatamab dose of 600 mg.

In both studies, a 24-h hospital stay was required following administration of study drugs at Cycle 1 Day 1 and also where a patient had experienced a grade ≥ 2 event of IRR/CRS at the previous administration. Patients were allowed to remain on therapy until loss of clinical benefit, PD, unacceptable toxicities, loss of cibisatamab exposure or withdrawal of consent. Investigators had the option to adjust the dose and/or the schedule of cibisatamab to allow patients who could potentially benefit from cibisatamab to remain on the study drug.

## Tumour response assessments

Tumour response was measured according to RECIST 1.1 and assessed by the investigators in S1 and by the investigators and independent review in S2.

Tumour assessment was performed once during screening. The first assessment after the start of treatment was performed at 8 weeks, and assessments were performed every 8 weeks thereafter for the first year and every 12 weeks thereafter until PD or treatment discontinuation.

## Safety

In both studies, the safety assessments were done at each visit and included vital signs, physical examinations, electrocardiograms, and laboratory tests. AEs were graded according to the NCI CTCAE v4.03, and CRS AEs were evaluated using NCI CTCAE v5.0. DLTs were defined as prespecified AEs that in S1 were related to cibisatamab and in S2 were related to cibisatamab or/and atezolizumab and in both cases had occurred during the DLT period (21 days following Cycle 1 Day 1) in patients enrolled during the dose-escalation part and during the step-up dose-escalation part of each study. Reported causality of the investigational drug was based on the judgment of the treating physicians. After each patient cohort (minimum of 3 patients) had been completed (i.e., the last patient enrolled in the cohort had reached Day 21), the Sponsor organized teleconferences with the investigators to discuss the safety and tolerability of cibisatamab (with or without obinutuzumab pretreatment) and in combination with atezolizumab and to determine the dose level for the next cohort. The next dose levels were recommended using a modified continual reassessment method with dose escalation overdose control (mCRM with EWOC) design (see Statistical Considerations) during the dose-escalation phase and discussed by the Sponsor and investigators. In addition, the clinical judgment of the Sponsor and investigators was also used in the dose-selection process.

## Pharmacokinetic and immunogenicity assessment

Serum PK and ADA samples were regularly collected. Validated bifunctional PK assays with a lower limit of quantification of 0.925 ng/mL (S1) and 2 ng/mL (S2) was used to determine cibisatamab serum concentrations. Anti-cibisatamab antibodies were analyzed in a three-tiered approach including screening, confirmation and titration assay using a validated method (bridging enzyme-linked immunosorbent assay). Patients were categorized being ADA-negative or ADA-positive as recommended by Shankar et al. [26]. Non-linear mixed effect modeling with software NONMEM version 7.4.3 (Icon Development Solutions, Ellicott City, Maryland) was used to analyze the dose-concentration-time data of cibisatamab.

## Pharmacodynamics and biomarker assessments

In S1 and S2, baseline and on-treatment mandatory paired biopsies were collected from all patients treated at dose levels >5 mg, except for patients with non-small cell lung cancer, for whom there was no accessible lesion. On-treatment tumour biopsy samples were assessed centrally by IHC for increases in intratumoural CD8 T cells (using PanCK-CD8 assay, Clone SP239 and Clone AE1/AE3/PCK26, at HistoGeneX), and PD-L1 (PD-L1 Clone SP263 at Ventana) expression on tumour cells and T cells. In addition, RNA sequencing was also performed on these matched-pair tissue samples (at Q2 Solutions) to assess changes at the gene level. Formalin-fixed paraffin-embedded (FFPE) tumour tissue was macrodissected to maximize tumour area by using hematoxylin and eosin as a guide. RNA extraction was conducted using High Pure FFPET RNA Isolation Kit (Roche) and assessed by Qubit and Agilent Bioanalyzer for quantity and quality. The prepared libraries were sequenced using the Illumina sequencing method. Whole blood, serum and plasma samples were regularly collected and processed centrally to analyse the number, activation and differentiation of immune cells by flow cytometry, to analyse soluble CEA as a disease-monitoring marker, to analyse cytokines and to assess relationships with dose levels, efficacy ADA status or AEs such as an IRR, or a CRS.

Additionally, we performed a retrospective subgroup analysis of efficacy by MSS-CRC status and of baseline *CEACAM5* gene expression in fresh baseline FFPE tumour tissue. Due to the limited dynamic range of the CEA IHC assay (using Cell Marque CEA31 monoclonal antibody, Catalog No. 236M-96, at Ventana), we analysed *CEACAM5* expression using RNA sequencing. Subgroup analyses of efficacy endpoints were performed based on a dichotomy of *CEACAM5* high (≥ 1500 RPKM) and low (< 1500 RPKM); 1500 RPKM was selected as the candidate biomarker cutoff based on the preliminary data, shown in Fig. 5A, resulting from the subgroup analyses.

## Study objectives

In S1, the primary objectives were to assess the safety of cibisatamab with or without obinutuzumab pretreatment in each cohort, as measured by the incidence of AEs and DLTs, to determine the MTD and/or the RP2D and schedule, to establish the pharmacokinetics of cibisatamab as monotherapy with or without obinutuzumab pretreatment and to assess the effect of obinutuzumab pretreatment on the rate and time of onset of ADA against cibisatamab. Secondary objectives were to evaluate preliminary anti-tumour activity of cibisatamab with or without obinutuzumab pretreatment and to describe the preliminary pharmacodynamic effects for cibisatamab with or without obinutuzumab pretreatment in mandatory paired tumour biopsies and paired blood samples.

In S2, the primary objectives were to establish the preliminary safety and tolerability profile, to determine the MTD in Cycle 1 and in later cycles, and to identify a RP2D of cibisatamab in combination with atezolizumab. Secondary objectives were to evaluate preliminary anti-tumour activity of cibisatamab in combination with atezolizumab and to describe the preliminary pharmacodynamic effects in mandatory paired tumour biopsies and paired blood samples.

## Statistical considerations

Studies S1 and S2 were analysed separately. The safety, efficacy, PK and pharmacodynamic analysis populations consisted of all patients who received at least one dose of any study drug. For efficacy analyses, at least one post-baseline measurement (such as a tumour assessment) was required for inclusion in the analysis.

To minimize the number of patients treated below therapeutically relevant doses, Part I of S1 consisted of single-patient cohorts for each dose level. In Part II of S1 and Part IA of S2, multiple patients were enrolled per dose level, and a mCRM with EWOC was used to determine the MTD; at least 3 patients per dose level were required for the mCRM. Dose-escalation decisions were made by the Sponsor and the participating investigators after the review of all collected relevant safety information. We also studied step-up dosing and QW vs Q3W schedules in S1 and S2.

Efficacy was reported descriptively. Binary endpoints, including ORR and DCR, were summarized using relative frequencies and 90% CI. Time-to-event endpoints, including progression-free survival, overall survival and duration of response, were summarized using Kaplan-Meier methodology as well as median time-to-event endpoints with 90% CI. Response endpoints were defined according to RECIST 1.1. Rave EDC (Medidata, New York, NY, USA) was used for capturing and managing clinical data. SAS version 9.4 (SAS Institute Inc., Cary, NC, USA) was used for statistical analyses of safety and efficacy data.

Analyses of biomarkers were post hoc and exploratory, with a focus on hypothesis generation. The Kruskal-Wallis rank sum test was used to assess the relationship between categorical and numerical variables, and the McNemar test was used to assess the relationship between two categorical variables.

## Reporting summary

Further information on research design is available in the Nature Portfolio Reporting Summary linked to this article.

## Data availability

Due to the informed consent provided by patients during trial enrolment, as well as applicable data protection laws in some countries, source data, including RNA sequencing data, cannot be shared publicly. However, qualified researchers can request it. Requests to access the data from the trials described in the current manuscript can be submitted through: https://vivli.org/members/enquiries-about-studies-not-listed-on-the-vivli-platform/. Requested data are available for 12 months from the approval of the request. For -up-to-date details on Roche's Global Policy on the Sharing of Clinical Information and how to request access to related clinical study documents, see here: https://go.roche.com/data_sharing. Anonymised records for individual patients across more than one data source external to Roche cannot, and should not, be linked due to a potential increase in risk of patient re-identification.

The remaining data are available within the Article and its Supplementary Information.

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

## Acknowledgements

The authors would like to thank the patients who participated in the trial, the patients' families, and the investigators and staff at all clinical study sites. This study was sponsored by F. Hoffmann-La Roche, Ltd. The sponsor played role in study design, data collection, data analysis and interpretation, and writing of the manuscript, in collaboration with the authors. Editorial assistance for this manuscript was provided by Jessica Bessler, PhD, CMPP of Nucleus Global, an Inizio Company, and funded by F. Hoffmann-La Roche, Ltd.

## Author contributions

Conceptualization: Neil H. Segal, Ignacio Melero, Victor Moreno, Neeltje Steeghs, Aurelien Marabelle, Daniel Waterkamp, Barbara Leutgeb, Said Bouseida, Nicholas Flinn, Markus Elze, Christopher Lieu, Emiliano Calvo, Josep Tabernero, Guillem Argilés Methodology: Neil H. Segal, Maria E. Rodriguez-Ruiz, Daniel Waterkamp, Barbara Leutgeb, Said Bouseida, Nicholas Flinn, Meghna Das Thakur, Hartmut Koeppen, Christopher Lieu, Josep Tabernero, Guillem Argilés Validation: Neil H. Segal, Cathy Eng, Markus Elze, Candice Jamois, Christopher Lieu, Emiliano Calvo Formal analysis: Neil H. Segal, Said Bouseida, Nicholas Flinn, Markus Elze, Candice Jamois, Christopher Lieu, Emiliano Calvo, Josep Tabernero Investigation: Neil H. Segal, Victor Moreno, Aurelien Marabelle, Neeltje Steeghs, Kristoffer Staal Rohrberg, Maria E. Rodriguez-Ruiz, Cathy Eng, Gulam Abbas Manji, Barbara Leutgeb, Said Bouseida, Christopher Lieu, Luis Paz-Ares, Josep Tabernero Resources: Neil H. Segal, Neeltje Steeghs, Kristoffer Staal Rohrberg, Joseph Paul Eder, Luis Paz-Ares, Josep Tabernero, Guillem Argilés Data curation: Neil H. Segal, N. Steeghs, Kristoffer Staal Rohrberg, Maria E. Rodriguez-Ruiz, Joseph Paul Eder, Gulam Abbas Manji, Said Bouseida, Nicholas Flinn, Candice Jamois, Christopher Lieu, Luis Paz-Ares, Josep Tabernero, Guillem Argilés Writing-original draft: Neil H. Segal, Ignacio Melero, Neeltje Steeghs, Daniel Waterkamp, Barbara Leutgeb, Said Bouseida, Nicholas Flinn, Meghna Das Thakur, Markus Elze, Hartmut Koeppen, Candice Jamois, Meret Martin-Facklam, Christopher Lieu, Emiliano Calvo Writing-review and editing: Neil H. Segal, Ignacio Melero, Victor Moreno, Aurelien Marabelle, Neeltje Steeghs, Kristoffer Staal Rohrberg, Maria E. Rodriguez-Ruiz, Cathy Eng, Gulam Abbas Manji, Daniel Waterkamp, Barbara Leutgeb, Said Bouseida, Nicholas Flinn, Meghna Das Thakur, Markus Elze, Hartmut Koeppen, Candice Jamois,,Meret Martin-Facklam, Christopher Lieu, Emiliano Calvo, Luis Paz-Ares, Josep Tabernero, Guillem Argilés Visualization: Neil H. Segal, Victor Moreno, Christopher Lieu, Supervision: Neil H. Segal, Aurelien Marabelle, Kristoffer Staal Rohrberg, Joseph Paul Eder, Markus Elze, Emiliano Calvo, Luis Paz-Ares, Josep Tabernero, Guillem Argilés.

## Competing interests

NHS reports research funding from Regeneron, Immunocore, Puretech, AstraZeneca, BMS, Merck, Pfizer, Roche/Genentech; served as an consultant/advisory board for Immunocore, PsiOxus, Roche/Genentech, Boehringer Ingelheim, Revitope, ABL Bio, Novartis, GSK, Astra Zeneca, Numab. IM reports grants from Roche, Bristol Myers, AstraZeneca, Genmab, Highlight therapeutics; consultancy fees from Roche, Bristol Myers, AstraZeneca, Genmab, Highlights therapeutics, Numab, F-Star, Catalym, Pharmamamar, Biolinerx, Gossamer, Bright peak, Pieris, Alligator, Pierre FabreVM reports consulting fees from: Roche, Bayer, BMS, Janssen and Basilea; Principal Investigator – Institutional Funding: AbbVie, AceaBio, Adaptimmune, ADC Therapeutics, Aduro, Agenus, Amcure, Amgen, Astellas, AstraZeneca Bayer Beigene BioInvent International AB, BMS, Boehringer, Boheringer, Boston, Celgene, Daichii Sankyo, DEBIOPHARM,Eisai, e-Terapeutics, Exelisis, Forma Therapeutics, Genmab, GSK, Harpoon, Hutchison, Immutep, Incyte, Inovio, Iovance, Janssen, Kyowa Kirin, Lilly, Loxo, MedSir, Menarini, Merck, Merus, Millennium, MSD, Nanobiotix, Nektar, Novartis, Odonate Therapeutics, Pfizer, Pharma Mar, PharmaMar, Principia, PsiOxus, Puma, Regeneron, Rigontec, Roche, Sanofi, Sierra Oncology, Synthon, Taiho, Takeda, Tesaro, Transgene, Turning Point Therapeutics, Upshersmith.. NS reports provided consultation or attended advisory boards for Boehringer Ingelheim, Ellipses Pharma; received research grants for the institute from AB Science, Abbvie, Actuate Therapeutics, ADCtherapeutics, Amgen, Array, Ascendis Pharma, Astex, AstraZeneca, Bayer, Blueprint Medicines, Boehringer Ingelheim, BridgeBio, Bristol-Myers Squibb, Cantargia, Celgene, CellCentric, Cresecendo, Cytovation, Deciphera, Eli Lilly, Exelixis, Genentech, Genmab, Gilead, GlaxoSmithKline, Incyte, InteRNA, Janssen/Johnson&Johnson, Kinate, Merck, Merck Sharp & Dohme, Merus, Molecular Partners, Novartis, Numab, Pfizer, Pierre Fabre, Regeneron, Roche, Sanofi, Seattle Genetics, Servier, Taiho, Takeda (outside the submitted work). AM over the last 5 years received honoraria and travel expense coverage for participation to Roche and Genentech Scientific Advisory Boards although not in direct relationship with the content of the manuscript. KSR reports compensation for conduction of clinical trial from Lilly, Roche/Genentech, Bristol-Myers Squibb, Symphogen, Pfizer, Novartis, Bayer, Alligator Bioscience, Incyte, Genmab, Puma Biotechnology, Orion Clinical, Bioinvent, Monta Bioscience, AstraZeneca; Speaker fees from Bayer, Amgen; Travel and conference reimbursement from AstraZeneca. MERR reports receiving research funding from Roche and Highlight Therapeutics. She also has received speaker's bureau honoraria from BMS and ROCHE. JE has nothing to disclose. CE reports research support for this trial provided to Vanderbilt-Ingram Cancer Center. GAM reports research funding from Regeneron, BioLineRx, Merck, Roche/Genentech; served on the advisory for CEND Pharmaceuticals, Roche/Genentech; owns stock in CEND Pharmaceuticals. DW is an employee of F. Hoffmann-La Roche. BL has nothing to disclose. SB owns stock in Roche. NF is an employee of F. Hoffmann-La Roche. MDT is an employee of F. Hoffmann-La Roche. ME is an employee of and owns stock in F. Hoffmann-La Roche AG. HK is an employee of Genentech, Inc. and owns stock in F. Hoffmann-La Roche, Ltd. CJ is an employee of and owns stock in F. Hoffmann-La Roche, Ltd. MMF reports Roche profit participation certificate. CL has nothing to disclose. EC reports payment or honoraria

from HM Hospitales Group; research funding form START; leadership role at start; consulting for Nanobiotix, Janssen-Cilag, Roche/Genentech, TargImmune Therapeutics, Bristol-Myers Squibb, Amunix, Adcendo, Anaveon, AstraZeneca/MedImmune, Chugai Pharma, MonTa, MDS Oncology, Nouscom, Novartis, Oncology, PharmaMar; employee of START, HM Hospitales Group; owns in START and Oncoart Associate; serves as president and found of foundation INTHEOS (Investigational Therapeutics in Oncological Sciences). LPA reports payment or honoraria from AstraZeneca, Janssen, Merck, Mirati; consulting fees from Lilly, MSD, Roche, Pharmamar, Merck, AstraZeneca, Novartis, Servier, Amgen, Pfizer, Sanofi, bayer, BMS, Mirati, GSK, Janssen, Takeda; research grants from MSD, AstraZeneca, Pfizer, BMS, member of board of directros for Altum Sequencing and Genomica; principle investigator for Alkermes, Amgen, AstraZeneca, Bristol Myers Squibb, Daiichi Sankyo, IO Biotech, Janssen-Cilag, Lilly, MSD, Novartis, Pfizer, Pharmamar, Roche, Sanofi, Takeda, and Tesaro. JT reports personal financial interest in form of scientific consultancy role for Array Biopharma, AstraZeneca, Avvinity, Bayer, Boehringer Ingelheim, Chugai, Daiichi Sankyo, F. Hoffmann-La Roche Ltd, Genentech Inc, HalioDX SAS, Hutchison MediPharma International, Ikena Oncology, Inspirna Inc, IQVIA, Lilly, Menarini, Merck Serono, Merus, MSD, Mirati, Neophore, Novartis, Ona Therapeutics, Orion Biotechnology, Peptomyc, Pfizer, Pierre Fabre, Samsung Bioepis, Sanofi, Seattle Genetics, Scandion Oncology, Servier, Sotio Biotech, Taiho,Tessa Therapeutics and TheraMyc. And also educational collaboration with Imedex, Medscape Education, MJH Life Sciences, PeerView Institute for Medical Education and Physicians Education Resource (PER); declares institutional financial interest in form of financial support for clinical trials or contracted research for Amgen Inc, Array Biopharma Inc,AstraZeneca Pharmaceuticals LP, BeiGene, Boehringer Ingelheim, Bristol Myers Squibb, Celgene, Debiopharm International SA, F. Hoffmann-La Roche Ltd, Genentech Inc, HalioDX SAS, Hutchison MediPharma International, Janssen-Cilag SA, MedImmune, Menarini, Merck Health KGAA, Merck Sharp & Dohme, Merus NV, Mirati, Novartis Farmacéutica SA, Pfizer, Pharma Mar, Sanofi Aventis Recherche & Développement, Servier, Taiho Pharma USA Inc, Spanish Association Against Cancer Scientific Foundation and Cancer Research UK. GA reports travel support from Amgen; speaker fees from Amgen, serves as medical advisor for Gadeta BV.

## Additional information

[1]Memorial Sloan Kettering Cancer Center, New York, NY, United States; Weill Cornell Medical College, New York, NY, USA. [2]Clínica Universidad de Navarra and CIMA University of Navarra, Navarra, Spain. [3]CIBERONC, Instituto de Salud Carlso III, Madrid, Spain. [4]Hospital Fundación Jiménez Díaz, Madrid, Spain. [5]Netherlands Cancer Institute, Amsterdam, Netherlands. [6]Gustave Roussy, Université Paris-Saclay, Villejuif, France. [7]Copenhagen University Hospital - Rigshospitalet, Copenhagen, Denmark. [8]Yale University Cancer Center, New Haven, CT, USA. [9]Vanderbilt Ingram Cancer Center, Nashville, TN, USA. [10]Columbia University, New York, NY, USA. [11]Genentech, Inc., South San Francisco, CA, USA. [12]F. Hoffmann-La Roche, Ltd, Basel, Switzerland. [13]University of Colorado, Aurora, Denver, CO, USA. [14]START Madrid-CIOCC, Centro Integral Oncológico Clara Campal, Madrid, Spain. [15]University Hospital 12 de Octubre, Madrid, Spain. [16]Vall d'Hebron Hospital Campus and Institute of Oncology (VHIO), Barcelona, Spain. [17]Departament de Cirurgia, Universitat Autònoma de Barcelona, Barcelona, Spain. ✉e-mail: segaln@mskcc.org

