## [Peer Review File · Nature Communications]

CEA-CD3 bispecific antibody cibisatamab with or without atezolizumab in patients with CEA-positive solid tumours: results of two multi-institutional Phase 1 trialsEditorial Note: Parts of this Peer Review File have been redacted as indicated to maintain patient confidentiality.

Reviewers' Comments:

Reviewer #1:

Remarks to the Author:

This manuscript provides a description of the development of cibusatamab, a CEA-CD3 bispecific antibody. Specifically, it reports on 2 trials, one using cibusatamab alone which enrolled 149 patients, the second in combination with atezolizumab which enrolled 228 patients. Subsets in each group were also treated with obinutuzumab.

The authors provide a concise and clear introduction to bispecific T-cell engager therapy and this agent in particular. They also provide descriptions of the safety profile fo cibusatamab,, both alone and in combination with atezolizumab and obinutuzumab, focusing on frequency and severity of adverse events. They described the pharmacokinetics and pharmacodynamics of the agent as well as its anti tumor effects. Finally, they attempted to describe biomarkers for response to this agent combination.

Several specific comments:

1.This is a novel agent and the trials explored it in depth answering several important questions. This manuscript is a distillation of a huge amount of work and data.

2.The major critique of this manuscript is that it is unclear why this large number (377) of patients was enrolled in the trials to arrive at these conclusions. The original study design called for single patient cohorts followed by multi-patient cohorts using a mCRM model with overdose control. This would appear to account for the 65 patient cohort receiving escalating doses of cibusatamab alone, in combination with atezolizumab (75 patients), and possibly the combination with obinutuzumab (27 patients). There is no description of the decision making that resulted in the step-up dose cohorts of the single agent or combination or the flat dose combination cohorts. The rationale for these additional study cohorts and the statistical considerations should be clearly explained.

3.Line 256: It is noted that 2 patients enrolled on study S2 (combined therapy with atezolizumab) had received prior therapy with cibusatamab. This seems unusual and should be explained.

4.Line 446: Although the authors note that the combination with atezolizumab showed no safety signal suggesting interacting toxicities, there is no comment as to toxicities from obinutuzumab therapy.

5. The biomarker results suggest that cibusatamab does act to increase T-cell infiltration via binding to CEA and CD3 as indicated in figures 8 and 9. The authors go on to suggest that CEACAM5 may be a biomarker response. However despite there statistical analysis suggesting significance in figure 7, the low numbers of responders and limited sampling make this speculative. They go on to suggest that baseline levels of CA expression may predict response to this approach. While logical this too is speculative based on only 2 responders.

6. In table A2, there appears to be some discrepancy in the number of patients assessed for response in the column headed "All step-Up". Including the nonresponders results are reported for 43 patients when the n=36

7. The section headed Summary And Conclusions should be shortened and focused.

Reviewer #2:

Remarks to the Author:

Dear authors and editors

Congratulations on this work combining two original prospective phase 1 dose escalation studies with a bi-specific antibody targeting CEA and CD3, with or without atezolizumab, in 377 patients with CEA-expressing cancers.

Protocols are well detailed, methodology and statistics are appropriate, dose escalations are specified, and toxicity, including specific toxicity, is very well detailed. At the flat dose level, the efficacy results (response rate about 15%) deserve further evaluation. The vast majority of patients have MSS colorectal cancer, and the results are specified in this population. Translational studies provide also further data for potential biomarkers.

These studies are highly relevant in the search for strategies for effective immunotherapies in MSS cancers.

This is a very high-quality manuscript.

Here are some minor comments:

In the abstract, one phase is incomplete: "In S1, 41 and S2, a total of 149 and 228 patients, respectively treatment".

Could you please detail the CEA tumor cell IHC expression assessment in CRC and non CRC (0, +, ++, +++)? Was there a central review?

Could serum CEA be a 1st line biomarker to select patients for further screening?

Reviewer #3:

Remarks to the Author:

This work is original and worth to be published. The data analysis is descriptive in general. Nonetheless, due to the multiple cohorts and designs, the interpretation of the results could be careful.

Overall, the methods are sound and the the results are informative. However, due to the multiple design and cohorts, it is hard to follow all details to reproduce the data analysis, although there are enough details included in the appendix. The authors did a great job to generate simple yet informative results from the two studies.

Reviewer #4:

Remarks to the Author:

The manuscript by Segal et al describes 2 clinical trials with cibisatamab, a CEA x CD3 bispecific alone (n=149) or in combination with atezolizumab (n=228). They found a high rate of adverse events (including 12 deaths) and antidrug antibodies (ADAs), but they also saw a modest response rate in colorectal cancer, which had been refractory to immunotherapy. They treated patients with Obinutuzumab to reduce ADA, which is relatively novel. The main figures are too numerous and could be easily consolidated, since most figures do not have many panels. The biomarker studies are fairly limited, with some IHC staining, RNAseq, and limited flow cytometry. The paper would benefit from reorganizing of the presentation to highlight the main points of the manuscript. The manuscript would also benefit from deeper analyses of the rich dataset from this large clinical trial.

1) The abstract is very terse with undefined abbreviations and lack of detail. The incidence of ADAs should be relayed there as should the incidence of ADAs with obinutuzumab. "In S1 and S2, a total of 149 and 228 patients, respectively treatment." Is an incomplete sentence.

2) With regards to the toxicity section, the MTD and DLT section should come after the "frequency and severity of AEs" section. The authors should provide some insights to the correlates with the development of IRR and CRS. Was this correlated with the levels of serum cytokines/CRP levels? Was this correlated with the levels of drug, levels of ADAs or antigen levels? These analyses should also be

included. What was the range of timing for CRS events with and without ADAs (Fig 1)?

3) The authors should be clear on what mitigation strategies (premedications, step-dosing) did to reduce the incidence of AEs, since this would have bearing on the paths forward. Did these reduce the levels of treatment induced cytokine levels? How did these affect the T cell kinetics?

4) One patient had a complete response (Table 4). This reviewer could not find details on what this case was.

5) For ADAs, this is reported as patients with and without ADAs (Fig 5). This figure is confusing: It is unclear what the 7 different groups in each panel are, or what the "n" is denoting.

6) The longitudinal PK and ADA data should be presented for this cohort. The impact of Obinutuzumab should be visually show in this context, rather than stating this in the results and in Table 3.

7) The authors try to make a point that this treatment can cause pseudoprogression with "inflammation" in Figures 3 and 4. The authors should also relay whether these cases when on to respond. Showing subsequent imaging would help illustrate the transient nature of this. These figures could be easily consolidated. Figure 8, showing the increase in T cells could also be consolidated with this to make the point of immune recruitment.

8) Part 1B randomized patients between weekly and every 3 week dosing. The authors should provide data and some conclusion from this. Were AEs, ADAs, responses,... different from these schedules? This is mentioned in passing (line 404). The mixed model analyses that were performed (line 408) should be presented rather than mentioned as "data on file".

9) The "Summary and Conclusions" section is on the one hand repetitive. This could be shortened, but could synthesize the results better.

10) They simply state (line 345) that the all 4 patient with high CEACAM5 levels by RNAseq had PRs. Were there any responses in the CEACAM5-low patients? How does doses and ADA factor in for these responses? CEACAM5 expression being associated with response is one of the conclusions of the paper, but this seems anecdotal without some form analysis.

11) The authors should provide some images of a clinical responder in addition to the spider plots.

12) The RNAseq data from pre and post-treatment biopsies is under analyzed and presented (Fig 10). The authors should present the data in there totality (heatmaps, volcano plots,...) that than just 2 gene scores.

13) The IHC analyses are also very simple. Figures 8 and 9 could be easily combined. Quantitation of the Ki67+ T Cells in the different compartments with statistical testing would strengthen this analyses.

14) The flow cytometry results presented are for CD4 and CD8 T cell counts (Fig A2). The authors should subset the T cells including looking at activation markers. Does the presence of ADA increase the frequency of circulating T cells as they speculate?

Reviewer #1:

The major critique of this manuscript is that it is unclear why this large number (377) of patients was enrolled in the trials to arrive at these conclusions. The original study design called for single patient cohorts followed by multi-patient cohorts using a mCRM model with overdose control. This would appear to account for the 65 patient cohort receiving escalating doses of cibusatamab alone, in combination with atezolizumab (75 patients), and possibly the combination with obinutuzumab (27 patients). There is no description of the decision making that resulted in the step-up dose cohorts of the single agent or combination or the flat dose combination cohorts. The rationale for these additional study cohorts and the statistical considerations should be clearly explained.

The main reason for the relatively high number of patients was due to expansive dose-schedule exploration introduced via several amendments, e.g. comparing 100 mg QW, 160 mg QW and 100 mg Q3W dosing in S2. In addition, various step-up dosing cohorts were explored in S1 and S2 to assess whether the approach of starting with low dose cibusatamab then dose-escalating would mitigate infusion-related reactions observed during early cycles. This has now been stated on page 5. Additionally, the demographic text has been rearranged for clarity (Pages 4 and 5).

Line 256: It is noted that 2 patients enrolled on study S2 (combined therapy with atezolizumab) had received prior therapy with cibusatamab. This seems unusual and should be explained. At the request of investigators and upon discussion with the study Medical Monitor, a small number of patients from S1 were allowed to participate in S2 after their participation in S1 was completed. Additional details are not available as we are not able to reliably identify these patients across study databases.

Line 446: Although the authors note that the combination with atezolizumab showed no safety signal suggesting interacting toxicities, there is no comment as to toxicities from obinutuzumab therapy.

An obinutuzumab safety summary is now included in the appendix (appendix page 7). AEs reported as related to obinutuzumab were in line with the known safety profile.

The biomarker results suggest that cibusatamab does act to increase T-cell infiltration via binding to CEA and CD3 as indicated in figures 8 and 9. The authors go on to suggest that CEACAM5 may be a biomarker response. However despite their statistical analysis suggesting significance in figure 7, the low numbers of responders and limited sampling make this speculative. They go on to suggest that baseline levels of CA expression may predict response to this approach. While logical this too is speculative based on only 2 responders.

Thanks for your comment. We agree with your assessment and removed the description of these anecdotal cases as well as figure 11.

In table A2, there appears to be some discrepancy in the number of patients assessed for response in the column headed "All step-Up". Including the nonresponders results are reported for 43 patients when the n=36

Thank you for spotting this. This was an error during the creation of the table and this has now been corrected.

The section headed Summary And Conclusions should be shortened and focused.

Thanks for your suggestion. The summary and conclusion sections have been revised and shortened.

Reviewer #2:

Here are some minor comments:

In the abstract, one phase is incomplete: "In S1, 41 and S2, a total of 149 and 228 patients, respectively treatment".

Thank you for finding this error. We have modified this abstract statement to read "In S1 and S2, a total of 149 and 228 patients, respectively, were enrolled.

Could you please detail the CEA tumor cell IHC expression assessment in CRC and non CRC (0, +, ++, +++)? Was there a central review?

As we have deleted figure 11 from the manuscript, this information is no longer needed.

Could serum CEA be a 1st line biomarker to select patients for further screening?

Thank you for the interesting suggestion. Unfortunately, this was not assessed during this study.

Reviewer #3:

This work is original and worth to be published. The data analysis is descriptive in general. Nonetheless, due to the multiple cohorts and designs, the interpretation of the results could be careful.

Overall, the methods are sound and the the results are informative. However, due to the multiple design and cohorts, it is hard to follow all details to reproduce the data analysis, although there are enough details included in the appendix. The authors did a great job to generate simple yet informative results from the two studies.

Reviewer #4:

The manuscript by Segal et al describes 2 clinical trials with cibisatamab, a CEA x CD3 bispecific alone (n=149) or in combination with atezolizumab (n=228). They found a high rate of adverse events (including 12 deaths) and antidrug antibodies (ADAs), but they also saw a modest response rate in colorectal cancer, which had been refractory to immunotherapy. They treated patients with Obinutuzumab to reduce ADA, which is relatively novel. The main figures are too numerous and could be easily consolidated, since most figures do not have many

panels. The biomarker studies are fairly limited, with some IHC staining, RNAseq, and limited flow cytometry. The paper would benefit from reorganizing of the presentation to highlight the main points of the manuscript. The manuscript would also benefit from deeper analyses of the rich dataset from this large clinical trial.

Thank you for your feedback. We have reorganized and focused the manuscript according to your and other reviewer's comments.

1) The abstract is very terse with undefined abbreviations and lack of detail. The incidence of ADAs should be relayed there as should the incidence of ADAs with obinutuzumab. "In S1 and S2, a total of 149 and 228 patients, respectively treatment." Is an incomplete sentence.

Thank you for finding this error. The incomplete sentence was completed to read:

"In S1 and S2, a total of 149 and 228 patients, respectively, were enrolled."

In addition, more information on ADA incidence was added to the abstract:

"In S1 and S2, 40 and 52% of patients, respectively, developed persistent anti-drug antibodies (ADAs). ADA-appearance could be mitigated by obinutuzumab-pretreatment with 8% of patients having persistent ADAs."

2) With regards to the toxicity section, the MTD and DLT section should come after the "frequency and severity of AEs" section. The authors should provide some insights to the correlates with the development of IRR and CRS. Was this correlated with the levels of serum cytokines/CRP levels? Was this correlated with the levels of drug, levels of ADAs or antigen levels? These analyses should also be included. What was the range of timing for CRS events with and without ADAs (Fig 1)?

Thank you for the comments and questions. We would like to leave the MTD/DLT section where it is as it provides a headline context for the rest of the section.

Additionally:

- In general, the release of cytokines IFN γ and IL-6 was dose-dependent and, as expected, events of IRR/CRS broadly correlated with increased cytokine levels. However, variability was large.
- At the first cibisatamab-infusion, comparable dose-dependent cytokine release was observed in ADA negative patients and those who went on to become ADA positive. The cytokine release observed following the first dose is expected based on the mechanism of action. After later infusions, cytokine release was higher in those patients who were ADA positive. The incidence of late cycle cytokine peaks (10-fold above 90th percentile of baseline distribution) was higher in ADA positive patients.
- Stoichiometry of ADA-titer and cibisatamab-concentration is indeed relevant for formation of the cibisatamab-ADA-T cell trimeric complex. A quantitative model explained well the occurrence and timing of the late cycle cytokine release. This model and results of a quantitative modeling analysis will be published in a separate publication.
- Onset of late cycle (ADA-mediated) CRS/IRR was early, i.e. during or shortly after infusion, whereas early cycle CRS (related to on-target tumor-related effects) typically occurred >6 hours after infusion.

3) The authors should be clear on what mitigation strategies (premedications, step-dosing) did to reduce the incidence of AEs, since this would have bearing on the paths forward. Did these reduce the levels of treatment induced cytokine levels? How did these affect the T cell kinetics? Thank you for the questions. The impact of corticosteroid premedication is difficult to quantify, since only patients enrolled in the initial dose escalation phases of S1 and S2 (i.e. lower doses of cibisatamab) did not receive corticosteroid premedication. All patients in step-up dosing cohorts and most patients treated with the highest doses of cibisatamab in S1 and S2, received corticosteroid prophylaxis. Corticosteroid premedication was also mandatory for patients who experienced a Grade ≥ 2 IRR in the previous Cycle. Hence the vast majority of patients treated with relevant therapeutic doses of cibisatamab received premedication.

The incidence of late cycle cytokine peaks (10-fold above 90th percentile of baseline distribution) tended to be more pronounced in step up dose cohorts probably due to repeated “favorable” stoichiometry for formation of cibisatamab-ADA-T cell trimeric complex, however sample size is small.

4) One patient had a complete response (Table 4). This reviewer could not find details on what this case was.

The patient with a complete response was a white female with MSI-H adenocarcinoma of the colon. She was enrolled in the 160 mg QW arm. We can include this information in the manuscript if the editor would like.

5) For ADAs, this is reported as patients with and without ADAs (Fig 5). This figure is confusing: It is unclear what the 7 different groups in each panel are, or what the “n” is denoting.

The data is by infusion. This was previously stated in the figure description, but is now also included in the figure itself. The n refers to the total number of patients receiving treatment at each administration. Data for the first 7 infusions are shown.

6) The longitudinal PK and ADA data should be presented for this cohort. The impact of Obinutuzumab should be visually show in this context, rather than stating this in the results and in Table 3.

Figure A3 showing the impact of obinutuzumab pretreatment on C_{min} and C_{max} has been added to the appendix. Additionally, the following sentence on page 8 “In the obinutuzumab-pretreated population, cibisatamab exposure was sustained in ADA-positive patients and the PK profiles were similar to those in ADA-negative patients” has been modified to “In the obinutuzumab-pretreated population, cibisatamab exposure was sustained in ADA-positive patients, and the cibisatamab concentration levels were similar to those in ADA-negative patients” to account for the new figure.

7) The authors try to make a point that this treatment can cause pseudoprogression with “inflammation” in Figures 3 and 4. The authors should also relay whether these cases when on

to respond. Showing subsequent imaging would help illustrate the transient nature of this. These figures could be easily consolidated. Figure 8, showing the increase in T cells could also be consolidated with this to make the point of immune recruitment.

Thank you for the suggestion. We have now added an end of treatment image to figure 3. We are not able to do so for figure 4 since that patient died and have therefore removed that figure.

8) Part 1B randomized patients between weekly and every 3 week dosing. The authors should provide data and some conclusion from this. Were AEs, ADAs, responses,... different from these schedules? This is mentioned in passing (line 404). The mixed model analyses that were performed (line 408) should be presented rather than mentioned as “data on file”.

While we appreciate the interest in the tumor growth kinetic modeling exploring impact of cibisatamab dose and schedule on shrinkage and growth rates. However, we prefer to describe the longitudinal model in a dedicated publication to allow sufficiently detailed information of model development and analyses.

9) The “Summary and Conclusions” section is on the one hand repetitive. This could be shortened, but could synthesize the results better.

Thanks for your suggestion. The summary and conclusion sections have been revised and shortened.

10) They simply state (line 345) that all 4 patients with high CEACAM5 levels by RNAseq had PRs. Were there any responses in the CEACAM5-low patients? How does dose and ADA factor in for these responses? CEACAM5 expression being associated with response is one of the conclusions of the paper, but this seems anecdotal without some form of analysis.

The details of the results by CEACAM5 levels are provided in figure 7. These data show that all of the responders seem to have a higher threshold of CEACAM5 by RNA. Here no PRs were seen in the low CEACAM5 patients. At this time we do not have further information around how dose and ADA factor into the responses.

11) The authors should provide some images of a clinical responder in addition to the spider plots.

The below figure shows computerized tomography scan images from a RECIST-confirmed partial responder with MSS mCRC taken at baseline and week 16 after receiving cibisatamab 160 mg QW and atezolizumab 1200 mg Q3W. These images have now been added to the manuscript as figure 5B.

12) The RNAseq data from pre and post-treatment biopsies is under analyzed and presented (Fig 10). The authors should present the data in there totality (heatmaps, volcano plots,...) that than just 2 gene scores.

Thank you for the suggestion. Unfortunately, we are not able to provide the requested analyses at this time.

13) The IHC analyses are also very simple. Figures 8 and 9 could be easily combined. Quantitation of the Ki67+ T Cells in the different comparments with statistical testing would strengthen this analyses.

Thank you for the suggestion. We have combined figures 8 and 9. Per the request for statistical testing, Ki67 is not an ideal marker to differentiate tumor from stroma for quantification of CD8 T cells in the tumor vs stromal compartments. Therefore, this comparison was not performed.

14) The flow cytometry results presented are for CD4 and CD8 T cell counts (Fig A2). The authors should subset the T cells including looking at activation markers. Does the presence of ADA increase the frequency of circulating T cells as they speculate?

We analyzed levels of activated, proliferating T cells in the periphery in the 100 and 160 mg cohorts of S2. Based on this data, it is not clear that there is a difference in the quantification of total T cells or activated T cells based on ADA status. However, we observed pronounced peaks of IL-6 in ADA positive vs. ADA negative patients especially in later cycles, which supports the cross-linking hypothesis and the ADA-driven activation of T cells.

Reviewers' Comments:

Reviewer #1:

Remarks to the Author:

The concerns raised in the initial review have been adequately addressed.

Reviewer #4:

Remarks to the Author:

In the revised manuscript by Segal et al, the authors address most of this reviewer's concerns, but some points are not fully addressed. The authors relay that a subsequent manuscript will focus on the specificity of the ADA as well as their mixed model analyses. The manuscript is improved with the revision, but still has items that should be addressed. Also, references within the manuscript is lacking for several figures.

Remaining points:

- 1) The author relay the institution of steroid pre-medications with the adoption of step-up dosing. The authors should provide detail on why this change was made (presumably because of CRS).
- 2) The authors now clarify that the one patient with a complete response was actually an MSI-H colorectal cancer patient. The paper emphasizes the responses in MSS colorectal cancer which were all partial responses. This review thinks that additional detail on this patient is important, since this was presumable the most dramatic response. Did the patient receive atezolizumab?
- 3) The authors did not perform any of the rudimentary analyses on the RNAseq data (prior point 12).
- 4) The authors present CEA expression data (Figure 6). Is the CEA expression indeed higher in the patients with partial responses. They must perform testing for statistical significance. This is one of the central points of the paper (highlighted in the abstract). If this is not significant, then this should be stated in the manuscript and removed from the abstract.
- 5) In figures 7B & D, the authors should perform testing for statistical significance. In the legend, "Ki57" should be corrected to "Ki67". There is minimal mention about the lack of association with clinical outcome (line 363). This should actually be discussed further, since it is clear that an increase in intratumoral T cells is not sufficient to induce clinical responses. Was the localization ("inflamed") associated with outcome. If not, this should also be clearly stated.
- 6) In figure 8, again the authors should perform testing for statistical significance.

Thank you for your further review of our manuscript entitled, "*CEA-CD3 bispecific antibody cibisatamab with or without atezolizumab in patients with CEA-positive solid tumours: results of two multi-institutional Phase 1 trials.*" Below we include our point-by-point responses to the reviewer comments. In a separate author checklist document, please find our responses to the editorial requests, as well as the journal policy and formatting requirements, which have largely been addressed.

Along with our submission, we also include a copy of the revised manuscript with changed text highlighted, production-quality figures, reporting summary and editorial policy checklist. We include a highlighted version of the revised supplementary information, as well as a final PDF version that is free of highlights and includes the redacted study protocols. Please also note that we have made slight updates to several tables in the Supplementary Information to ensure accuracy. These do not impact any of the conclusions from the manuscript. Please also see our responses in the author checklist regarding data sharing requests.

I would be happy to discuss anything further with you as needed. We hope that with these revisions, our submission will now be suitable for publication in *Nature Communications*

RESPONSES TO REVIEWERS' COMMENTS

Reviewer #1 (Remarks to the Author):

The concerns raised in the initial review have been adequately addressed.

>>We thank the reviewer for their additional review of our manuscript.

Reviewer #4 (Remarks to the Author):

In the revised manuscript by Segal et al, the authors address most of this reviewer's concerns, but some points are not fully addressed. The authors relay that a subsequent manuscript will focus on the specificity of the ADA as well as their mixed model analyses. The manuscript is improved with the revision, but still has items that should be addressed. Also, references within the manuscript is lacking for several figures.

Remaining points:

1) The author relay the institution of steroid pre-medications with the adoption of step-up dosing. The authors should provide detail on why this change was made (presumably because of CRS).

>>As stated in the revised manuscript, step-up dosing cohorts were explored to assess whether the approach of starting with low dose cibisatamab and then dose-escalating would mitigate infusion-related reactions. A correlation between frequency and severity of early-cycle adverse events and dose-levels was observed in the flat dose cohorts. Therefore, it was hypothesized that using a lower dose in early cycles and increasing this in later cycles would improve the safety profile while potentially not impacting efficacy. Additionally, it was explored whether using a higher dose in later cycles could overcome the impact of ADAs on exposure. This approach was not pursued further as it may have increased the severity of late-cycle CRS and was not successful in overcoming the exposure issue.

2) The authors now clarify that the one patient with a complete response was actually an MSI-H colorectal cancer patient. The paper emphasizes the responses in MSS colorectal cancer which were all partial responses. This review thinks that additional detail on this patient is important, since this was presumably the most dramatic response. Did the patient receive atezolizumab?

>>Yes, this patient was enrolled into the combination arm and received 1200 mg atezolizumab Q3W in combination with 160 mg cibusatamab QW (flat dosing). This was [redacted] with adenocarcinoma of the colon. As per Roche clinical database, the patient received prior FOLFOX chemotherapy and no prior checkpoint inhibitor. The patient developed persistent ADAs, which further supports the hypothesis that the CR was likely driven by atezolizumab.

3) The authors did not perform any of the rudimentary analyses on the RNAseq data (prior point 12).

>>The provided data is purely descriptive and hypothesis generating. Unfortunately, the requested additional analyses are not available at this time. Currently there is no plan to perform these on the available small sample. Instead, larger future datasets generated on novel TCBs may allow for more robust sub-analyses.

4) The authors present CEA expression data (Figure 6). Is the CEA expression indeed higher in the patients with partial responses. They must perform testing for statistical significance. This is one of the central points of the paper (highlighted in the abstract). If this is not significant, then this should be stated in the manuscript and removed from the abstract.

>>This was an exploratory analysis and therefore purely hypothesis generating. We have not provided a p-value to avoid giving the reader a false sense of confidence in a result that was not based on pre-specified criteria. Factors such as trying a number of hypotheses, only having selected patients with data available, as well as using different statistical tests can lead to overly optimistic p-values. The chi-squared test p-value is 0.00387. If you still think it is important to include this p-value, we suggest to revise the sentence in the Results section as follows:

Additionally, efficacy seemed to be further enhanced in patients with high CEACAM5 expression (Figure 6, chi-squared test $p=0.00387$, caveat: exploratory post-hoc analysis).

5) In figures 7B & D, the authors should perform testing for statistical significance. In the legend, "Ki57" should be corrected to "Ki67". There is minimal mention about the lack of association with clinical outcome (line 363). This should actually be discussed further, since it is clear that an increase in intratumoral T cells is not sufficient to induce clinical responses. Was the localization ("inflamed") associated with outcome. If not, this should also be clearly stated.

>>Thanks for spotting this error, Ki57 should be corrected to Ki67. We did not further comment on this finding due to the limited word-count and to avoid speculation. Further data will need to be generated to better understand the lack of association, but it is apparent that increased CD8 infiltration is not always sufficient to induce clinical responses. The quality of the immune infiltrate, T cell exhaustion and/or lack of co-stimulation (signal 2) may be contributing factors.

6) In figure 8, again the authors should perform testing for statistical significance.

>>The comparison of antigen-processing and T-effector gene sets is exploratory and meant to be purely descriptive. At $n=23$ the samples are highly selected compared to the overall trial population, so there is a risk of bias. The point of Figure 8 is to support the drug mechanism of action, but without making any strong claims based on the results. Tests such as a t test produce significant p-values, but we do not recommend including them here to avoid giving the reader too much confidence in these results.